# Spatial Conformal Inference through Localized Quantile Regression

**Hanyang Jiang** [1]   **Yao Xie** [1]

## Abstract

Reliable uncertainty quantification at unobserved spatial locations is a key challenge in spatial statistics, particularly for complex and heterogeneous datasets. While traditional methods such as Kriging rely on strong distributional assumptions, conformal prediction (CP) offers a distribution-free alternative. However, although non-i.i.d. CP theory is well established for time-series data, a significant gap remains for spatial data, where the lack of a natural ordering and discrete index complicates theoretical guarantees. Existing CP theory for spatial data often relies on exchangeability. We propose *Localized Spatial Conformal Prediction* (LSCP), a model-agnostic framework that bridges this gap by coupling local quantile regression with conformal calibration. LSCP conditions on spatial neighborhoods to capture local heterogeneity. We show that LSCP retains finite-sample marginal coverage under spatial exchangeability and attains asymptotic conditional coverage under stationarity and spatial mixing. Across synthetic and real datasets, LSCP consistently achieves near-nominal coverage with tighter and more stable prediction intervals than existing methods that fail to capture these spatial dependencies.

## 1. Introduction

Quantifying uncertainty at unobserved spatial locations has long been a central challenge in spatial statistics, particularly in applications such as weather forecasting (Siddique et al., 2022) and mobile signal coverage estimation (Jiang et al., 2024). Traditional methods such as Kriging rely on strong parametric assumptions, including normality and stationarity, to model spatial relationships and quantify uncertainty (Cressie, 2015). However, violations of these assumptions in complex spatial datasets (Heaton et al., 2019) often lead to unreliable uncertainty quantification. This issue is especially pronounced for prediction intervals, whose validity can be severely compromised by non-stationarity or departures from Gaussianity (Fuglstad et al., 2015).

While a broad literature has addressed spatial heterogeneity (Gelfand et al., 2005; Duan et al., 2007), many existing approaches are computationally expensive and do not scale well to large datasets. Moreover, fully specifying a generative spatial model is not always necessary when the primary goal is reliable uncertainty quantification rather than likelihood-based inference. In parallel, machine learning methods have been increasingly used for spatial prediction (Hengl et al., 2018; Chen et al., 2024), but they typically focus on point prediction and often lack uncertainty estimates with formal coverage guarantees.

Conformal prediction (CP), introduced by (Vovk et al., 2005), provides a distribution-free framework for uncertainty quantification. By leveraging exchangeability, CP produces prediction sets with finite-sample marginal coverage without assumptions on the data distribution or the prediction model (Lei & Wasserman, 2014; Angelopoulos et al., 2023). This property makes CP attractive in settings involving black-box predictors or complex data-generating processes.

However, in many real-world datasets, including time-series and spatial data, the assumption of exchangeability does not hold. Recent work has therefore extended CP to non-exchangeable settings. For example, (Tibshirani et al., 2019) proposed weighted conformal methods to address covariate shift by reweighting calibration samples, while (Barber et al., 2023) analyzed coverage gaps under general distributional shifts. For time-series data, further advances have established tighter prediction intervals and asymptotic guarantees under temporal dependence (Xu & Xie, 2023; Xu et al., 2024).

In this paper, we extend conformal prediction to the spatial domain, where non-exchangeability arises from multidimensional dependence and the absence of a natural ordering. While time series can be viewed as a one-dimensional special case of spatial data, spatial locations are continuous and lack a canonical index, making direct extensions of time-series CP theory inadequate. Existing CP methods

[1]H. Milton Stewart School of Industrial and Systems Engineering, Georgia Institute of Technology, Atlanta, Georgia, USA. Correspondence to: Hanyang Jiang <scottjhy@gatech.edu>, Yao Xie <yao.xie@isye.gatech.edu>.

*Proceedings of the 43rd International Conference on Machine Learning*, Seoul, South Korea. PMLR 306, 2026. Copyright 2026 by the author(s).

for spatial data often rely on exchangeability, limiting their applicability in heterogeneous settings. To address this gap, we propose *Localized Spatial Conformal Prediction* (LSCP), a framework that couples localized quantile regression with conformal calibration to construct prediction intervals under general spatial mixing conditions. Our main contributions are summarized as follows:

- *Localized Spatial Conformal Prediction (LSCP):* We introduce LSCP, a conformal prediction method tailored to spatial data, which learns data-adaptive localization through quantile regression on neighborhood residuals and adapts to complex spatial dependence.

- *Theoretical guarantees:* We establish finite-sample bounds on the coverage gap and asymptotic conditional coverage under stationarity and spatial mixing, without requiring exchangeability. When exchangeability holds, LSCP recovers the standard distribution-free marginal coverage guarantee of classical CP.

- *Empirical evaluation:* We evaluate LSCP on synthetic and real-world datasets and show that it achieves tighter prediction intervals with valid coverage and more uniform calibration across the spatial domain.

## 1.1. Literature

*Conformal prediction beyond exchangeability.* Traditional conformal prediction relies on exchangeability, which is often violated in real-world settings such as time series forecasting. Recent work has extended conformal prediction to non-exchangeable settings, with a prominent line based on weighted conformal prediction. For example, Tibshirani et al. (2019) addressed covariate shift by weighting samples using likelihood ratios, and Barber et al. (2023) provided a general framework that bounds coverage gaps under distribution shift, including bounds based on total variation distance. In the time series setting, methods often assign higher weights to recent data (Xu et al., 2024) or adapt the significance level $\alpha$ online to track distribution drift (Gibbs & Candes, 2021; Angelopoulos et al., 2024). However, finding a universally optimal weighting strategy remains open (Barber et al., 2023). In the absence of exchangeability, finite-sample guarantees are typically replaced by asymptotic validity under additional dependence, mixing, or stability conditions.

*Conformal prediction for spatial and localized data.* The spatial setting is broader and more complex than time series data, yet conformal methods tailored to spatial contexts remain relatively limited. A recent study by (Mao et al., 2024) introduced a spatial conformal prediction method under an infill sampling framework, using kernel functions to weight conformity scores based on spatial proximity. This aligns

with a broader class of localized conformal methods that aim to approximate conditional validity. For instance, (Guan, 2023) and (Han et al., 2022) proposed general frameworks for localized conformal prediction, using kernel functions to adapt prediction intervals to the local feature or covariate space. More recently, theoretical work has formalized conditional objectives in non-exchangeable settings: (Gibbs et al., 2025) studied conditional guarantees under distribution shift, while (Colombo, 2024) and (Plassier et al., 2024) used generative models and conditional density estimation to construct locally adaptive prediction sets with approximate conditional validity. While not all of these methods are designed specifically for spatial data, they highlight the role of local weighting and conditional estimation in handling complex dependence and heterogeneity.

## 2. Problem setup

In this paper, we consider a spatial setting with observations $\{Z(s_i)\}_{i=1}^n$, where $Z(s) = (X(s), Y(s))$ represents a random field observed at spatial locations $s_i$. Here, $Y(s) \in \mathbb{R}$ is the response variable, and $X(s) \in \mathbb{R}^p$ is the associated feature vector, potentially including the spatial location $s$. In this paper, we follow the commonly used stochastic design in spatial settings, meaning that the spatial locations $s_i$ are i.i.d. samples from an unknown distribution $g(s)$.

In conformal prediction, the objective is to construct a prediction region $\hat{C}_n(X)$ for an unobserved response $Y$ given a known feature vector $X$. For a user-specified confidence level $\alpha$, we aim to ensure that the probability of $Y$ falling within the prediction region is at least $1 - \alpha$. This notion of coverage can be interpreted in two ways: marginal coverage and conditional coverage. *Marginal coverage* is defined as

$$\mathbb{P}(Y \in \hat{C}_n(X)) \geq 1 - \alpha,$$

whereas *conditional coverage* requires that

$$\mathbb{P}(Y \in \hat{C}_n(X) \mid X = x) \geq 1 - \alpha.$$

Conditional coverage is stricter than marginal coverage, requiring validity for all values of $X$. However, as shown by (Foygel Barber et al., 2021), achieving universal conditional coverage is impossible without additional assumptions. In standard conformal settings, where the data are i.i.d. or exchangeable, only marginal coverage is typically guaranteed.

## 3. Method

We first provide the necessary background on the conventional split conformal prediction framework, which yields prediction intervals of uniform width. Building upon this, we introduce our proposed Localized Spatial Conformal Prediction (LSCP) method, designed to generate more adaptive intervals by modeling spatial correlation. We situate our

$$(\varepsilon_1, \cdots, \varepsilon_n) \overset{d}{=} (\varepsilon_{\sigma(1)}, \cdots, \varepsilon_{\sigma(n)})$$

Exchangeability

$$(\varepsilon(s_1), \cdots, \varepsilon(s_n)) \overset{d}{=} (\varepsilon(s_1 + \delta), \cdots, \varepsilon(s_n + \delta)), \forall \delta$$

Spatial Stationarity

*Figure 1.* Difference between exchangeability and spatial stationarity. Exchangeability means permutation-invariant while stationarity represents translation-invariant.

contribution by comparing it with several related techniques in the recent literature: Global Spatial Conformal Prediction (GSCP), Smoothed Localized Spatial Conformal Prediction (SLSCP) from (Mao et al., 2024), and Localized Conformal Prediction (LCP) from (Guan, 2023). A detailed discussion is provided in Appendix B, and Table 1 summarizes the differences.

### 3.1. Background: Conformal prediction

Conformal prediction constructs a prediction interval for $Y_{n+1}$ given a prediction model $\hat{f}$, the feature vector $X_{n+1}$, and past observations $\{(X_i, Y_i)\}_{i=1}^n$. The prediction model $\hat{f}$ can be any user-specified model. The data are split into a training set to fit $\hat{f}$ and a calibration set to compute non-conformity scores, typically $\hat{\varepsilon}_i = |Y_i - \hat{f}(X_i)|$. Using the empirical quantile $\widehat{Q}_n(1 - \alpha)$ of these scores, the prediction interval is constructed as

$$\hat{C}_n(X_{n+1}) = [\hat{f}(X_{n+1}) - \widehat{Q}_n(1-\alpha), \ \hat{f}(X_{n+1}) + \widehat{Q}_n(1-\alpha)].$$

Conformal prediction ensures valid marginal coverage of $1 - \alpha$ under exchangeability and is flexible, requiring no assumptions on the distribution of $Y$ or the form of $\hat{f}$.

In contrast, traditional geospatial methods such as Gaussian Processes (GPs) assume a Gaussian prior with an explicit covariance structure, providing mean predictions and uncertainty estimates. While GPs are powerful, their reliance on Gaussian assumptions can lead to poor coverage when data deviate from this distribution. GPs are also computationally expensive, limiting scalability to large datasets. Conformal prediction, by contrast, is model-agnostic and provides finite-sample coverage guarantees, making it practical in many scenarios.

### 3.2. Local spatial conformal prediction (LSCP)

We propose the LSCP algorithm. In spatial settings, data often exhibit significant dependence across locations, and taking the spatial dependence into account can greatly improve the accuracy of prediction intervals. To account for this, it is advantageous to base predictions on nearby data points, as spatially proximate observations are likely to share similar distributions and therefore provide more reliable information than distant, potentially uncorrelated samples. The recent study by (Barber et al., 2023) highlights the importance of weighting calibration data differently, depending

on their relevance to the target prediction point. By assigning higher mass to local samples, this approach effectively creates a "virtual" calibration set that mimics the test distribution, mitigating the distribution shift inherent in spatial data.

Assume the calibration set consists of observations $(X(s_1), Y(s_1)), \ldots, (X(s_n), Y(s_n))$. For a new observation $(X(s_{n+1}), Y(s_{n+1}))$, the aim is to construct a prediction interval by selecting a neighborhood of data from the calibration data. Here we use $N(s_{n+1})$ to represent the neighborhood of $s_{n+1}$, which can be determined via various criteria, and a common approach is to include all nearby points located within a specified distance threshold. In the paper, we use $k$-nearest neighbors for simplicity.

Given a pre-trained prediction model $\hat{f}$, the non-conformity scores are defined as $\hat{\varepsilon}(s_i) = Y(s_i) - \hat{f}(X(s_i))$. Let $\mathcal{E}_{n+1} = \{\hat{\varepsilon}(s_i)\}_{i \in N(s_{n+1})}$ be the non-conformity scores of the neighbors of $s_{n+1}$. We then define the conditional cumulative distribution function (CDF), denoted by

$$F(e \mid \mathcal{E}_{n+1}) = \mathbb{P}(\hat{\varepsilon}(s_{n+1}) \le e \mid \mathcal{E}_{n+1}).$$

The conditional quantile $Q_n(p)$ is defined as

$$Q_n(p) = \inf\{e \in \mathbb{R} : F(e \mid \mathcal{E}_{n+1}) \ge p\}. \tag{1}$$

Let $\widehat{Q}_n(p)$ be an estimator of the true quantile $Q_n(p)$ in Equation 1; the prediction interval of LSCP is defined as:

$$\hat{C}_n(X(s_{n+1})) = [\hat{f}(X(s_{n+1})) + \widehat{Q}_n(\beta^*),$$
$$\hat{f}(X(s_{n+1})) + \widehat{Q}_n(1 - \alpha + \beta^*)],$$

where $\beta^* = \operatorname{argmin}_{\beta \in [0,\alpha]}(\widehat{Q}_n(1 - \alpha + \beta) - \widehat{Q}_n(\beta))$. Here $\beta$ is optimized to find the tightest interval. In particular, if we choose $\widehat{Q}_n(p)$ to be the empirical quantile and $\beta^* = \alpha/2$, then LSCP reduces to a localized version of GSCP.

In order to leverage the dependency in residuals and produce intervals as narrow as possible, we apply a quantile regression estimator $\widehat{Q}_n$ to predict the conditional quantile of the residual at $s_{n+1}$ based on its neighboring residuals $\{\hat{\varepsilon}(s_i)\}_{i \in N(s_{n+1})}$. For computational efficiency, we use Quantile Random Forests (QRF) from (Meinshausen, 2006), although other quantile regression techniques could also be applied. The QRF learns the conditional quantile by treating the residual at location $s_{n+1}$, denoted $\tilde{Y}(s_{n+1}) = \hat{\varepsilon}(s_{n+1})$, as a function of the vector of its neighboring residuals, $\tilde{X}(s_{n+1}) = (\hat{\varepsilon}(s_i))_{i \in N(s_{n+1})}$. Here we order the residuals by increasing spatial distance $\|s_i - s_{n+1}\|$ to preserve the local dependence structure. This consistency is critical for leveraging the efficiency of QRF. The estimator yielded by the QRF is a weighted empirical quantile:

$$\widehat{Q}_n(p) = \inf\{e \in \mathbb{R} : \sum_{i=1}^n \omega_i \mathbf{1}\{\tilde{Y}(s_i) \le e\} \ge p\},$$

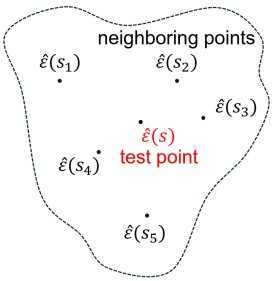
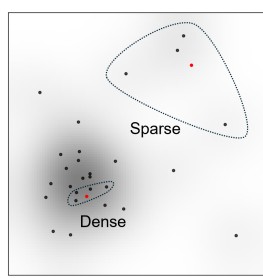

*Figure 2.* Illustration of neighborhoods in Localized Spatial Conformal Prediction (LSCP). Left: $k$-nearest neighbors are used as features to capture spatial dependence. Right: neighborhoods adapt to both dense and sparse regions. Red points denote test locations, dashed circles indicate the 5-nearest neighbors, and darker shading corresponds to lower uncertainty.

---

**Algorithm 1** Localized Spatial Conformal Prediction

**Require:** Dataset $\{(x(s_i), y(s_i))\}_{i=1}^n$, prediction algorithm $\mathcal{A}$, significance level $\alpha$

**Ensure:** Prediction intervals $\{\widehat{C}_n(x(s_{n+1}))\}$

1: Split the dataset into training data and calibration data.
2: Train the prediction model $\hat{f}$ with the training data using the prediction algorithm $\mathcal{A}$.
3: Construct neighborhoods $N(s_i)$ for all calibration points and $N(s_{n+1})$ for the test point.
4: Compute the non-conformity scores $\hat{\varepsilon}(s_i) = Y(s_i) - \hat{f}(X(s_i))$ for all data in the calibration set.
5: Set $\tilde{Y}(s_i) = \hat{\varepsilon}(s_i)$ and $\tilde{X}(s_i) = (\hat{\varepsilon}(s_{j_1}), \ldots, \hat{\varepsilon}(s_{j_{|N(s_i)|}}))$, where $s_j \in N(s_i)$.
6: Fit quantile regression $\widehat{Q}_n$ with all pairs $(\tilde{X}(s_i), \tilde{Y}(s_i))$ in the calibration data.
7: Obtain the prediction interval $\hat{C}_n(X(s_{n+1}))$.

---

where the weights $\omega_i$ are learned through quantile regression. The derivation is detailed in Appendix A.1. One key difference between LSCP and other weighted spatial conformal prediction methods lies in the fact that the weights $\omega_i$ are data-adaptive rather than user-specified, and Table 1 summarizes the differences.

### 3.3. Computational considerations and scalability

LSCP is computationally efficient and scalable to large spatial datasets. It fits a single global quantile regression model on the calibration data and performs a $k$-nearest neighbor search followed by one forward evaluation at test time. Since the neighborhood size $k$ is small and fixed, both training and inference scale favorably, providing an efficient alternative to kernel-weighted or fully localized conformal methods. A more detailed analysis is in Appendix C.5.

## 4. Theoretical results

In this section, we establish the theoretical foundations of the proposed LSCP framework. Our primary goal is to demonstrate that LSCP provides valid coverage guarantees even when the standard assumption of exchangeability is violated due to spatial dependence. We begin in Section 4.1 by defining the problem setup and necessary notations. We then introduce the concept of spatial mixing in Section 4.2 to rigorously quantify the dependence structure of the random field. Building on these definitions, we outline the key assumptions required for our analysis in Section 4.3. Finally, we present our main theoretical results: while finite-sample marginal coverage holds under the simplified assumption of exchangeability (Section 4.4), our central contribution is the derivation of asymptotic conditional coverage guarantees under general stationary and spatially mixing conditions (Section 4.5).

### 4.1. Notations

Suppose the data are denoted by $\{Z(s_i)\}_{i=1}^n$, where $Z(s) = (X(s), Y(s))$, $s \in \mathbb{R}^d$ denotes the spatial location, $X(s) \in \mathbb{R}^d$ is the feature vector, and $Y(s) \in \mathbb{R}$ represents the univariate response. We assume that $Y(s)$ is generated from a true model with unknown additive noise:

$$Y(s) = f(X(s)) + \varepsilon(s),$$

where $f$ is an unknown function and $\varepsilon(s)$ represents the noise process, whose marginal distribution is not necessarily Gaussian. Given a pre-trained prediction model $\hat{f}$, we can compute the non-conformity scores

$$\hat{\varepsilon}(s) = Y(s) - \hat{f}(X(s)).$$

The estimated empirical CDF $\widehat{F}_{n+1}(y)$ is defined as

$$\widehat{F}_{n+1}(y) = \sum_{i=1}^n \omega_i \mathbf{1}(\hat{\varepsilon}(s_i) \le y),$$

where $\omega_i$ is the weight assigned to data point $s_i$ and satisfies $\sum_{i=1}^n \omega_i = 1$. Besides, we define the CDF and the weighted empirical CDF of the true noise as $F_\varepsilon(y)$ and $\widetilde{F}_{n+1}(y)$, respectively, where

$$\widetilde{F}_{n+1}(y) = \sum_{i=1}^n \omega_i \mathbf{1}(\varepsilon(s_i) \le y).$$

### 4.2. Preliminary

*Spatial mixing.* To derive theoretical guarantees, we must quantify the extent to which the random field $Z(\cdot)$ exhibits dependence, that is, how quickly variables become statistically independent as the distance between them increases.

*Table 1.* Comparison of assumptions and algorithms across localized conformal prediction methods.

| | **LSCP** (Ours) | **SLSCP** (Mao et al., 2024) | **LCP** (Guan, 2023) |
|---|---|---|---|
| Algorithm | Weighted quantile via learned quantile regression | Weighted quantile via fixed kernel function | Weighted quantile via fixed kernel function |
| Weights | Data-adaptive, learned by quantile regression on residual features | User-specified kernel based on spatial distance | User-specified kernel based on feature similarity |
| Dependence captured | Complex dependence learned directly from local residual patterns | Spatial dependence limited to pairwise proximity via a given kernel | Feature dependence limited to pairwise similarity via a given kernel |
| Distributional assumptions | Stationary, spatially mixing noise | Infill sampling with locally i.i.d. noise | Globally i.i.d. data |
| Data model | Additive noise model | Continuous mapping from separate spatial and noise processes | No explicit structural assumption, but requires i.i.d. data |

First, we define a measure of dependence between two specific spatial regions. Let $\mathcal{F}_Z(T) = \sigma\langle Z(s) : s \in T\rangle$ denote the $\sigma$-algebra generated by the random field on a subset $T \subset \mathbb{R}^d$. For any two disjoint regions $T_1, T_2 \subset \mathbb{R}^d$, the strong mixing coefficient between them is defined as the maximum difference between the joint probability and the product of marginal probabilities:

$$\tilde{\alpha}(T_1, T_2) = \sup\{|\mathbb{P}(A \cap B) - \mathbb{P}(A)\mathbb{P}(B)| : $$
$$A \in \mathcal{F}_Z(T_1),\ B \in \mathcal{F}_Z(T_2)\}.$$

Intuitively, $\tilde{\alpha}(T_1, T_2)$ measures the strongest dependence between any event occurring in region $T_1$ and any event in region $T_2$.

Next, we define the mixing coefficient for the entire process. Unlike time series ($d = 1$), where past and future are clearly defined half-lines, spatial sets can take complex shapes. To handle this, we adopt the framework established by (Lahiri, 2003), which defines the mixing coefficient $\alpha(a; b)$ based on the worst-case dependence between sets separated by a distance $a$ with volume bounded by $b$.

Let $d(T_1, T_2) = \inf\{|x - s| : x \in T_1,\ s \in T_2\}$ be the minimum distance between two sets. We restrict our attention to $\mathcal{R}_k(b)$, the collection of all sets formed by the union of at most $k$ disjoint cubes in $\mathbb{R}^d$ with total volume bounded by $b$:

$$\mathcal{R}_k(b) \equiv \left\{\cup_{i=1}^k D_i : \sum_{i=1}^k |D_i| \le b\right\}.$$

The strong mixing coefficient $\alpha(a; b)$ for the random field is then defined as:

$$\alpha(a; b) = \sup\left\{\tilde{\alpha}(T_1, T_2) : d(T_1, T_2) \ge a,\ T_1, T_2 \in \mathcal{R}_3(b)\right\}.$$

Here, the parameter $a$ controls the *separation distance*, and $b$ controls the *volume* of the regions.

Finally, to ensure valid coverage, we assume that this coefficient satisfies a standard decay condition: it decays poly-nomially with distance while growing at most polynomially with volume. Specifically, we posit the existence of a non-increasing function $\alpha_1(\cdot)$ with $\lim_{a\to\infty} \alpha_1(a) = 0$ and a non-decreasing function $g(\cdot)$ such that:

$$\alpha(a; b) \le \alpha_1(a)g(b), \quad \text{for all } a > 0, b > 0.$$

### 4.3. Assumptions

With the notation above, we now state the assumptions that underlie our theoretical analysis. When the data are exchangeable, valid marginal coverage follows directly when using the empirical quantile as the quantile estimate. Our focus, however, is the non-exchangeable setting, which is common under localization and spatial dependence, where we replace exchangeability with stationarity and spatial mixing to establish coverage guarantees.

**Assumption 4.1** (Weight decay)**.** There exists a constant $\gamma > 0$ such that the normalized weights satisfy

$$\omega_i = o\left(n^{-\frac{1+\gamma}{2}}\right), \tag{2}$$

for all $i$, meaning that $M_n = \max_{1 \le i \le n} \omega_i = o\left(n^{-\frac{1+\gamma}{2}}\right)$.

The requirement assumes that the normalized weights decay at a rate faster than $n^{-\frac{1}{2}}$. We emphasize that this assumption does not require the neighborhood size $k$ to grow, since the QRF is trained on the whole calibration dataset. As we can see, $\omega_i = \frac{1}{n}$ is a special case that satisfies this condition. The condition is also weaker than the requirement of $\omega_i = O(\frac{1}{n})$ in a related study (Xu et al., 2024). Besides, the assumption can also be inferred from that of SLSCP (Mao et al., 2024), where a GBF kernel with infinitely close data leads to uniform weights. Moreover, the assumption automatically holds for QRF when setting the minimum samples per leaf to be $\lceil cn^\eta \rceil$ for constants $c > 0$ and $\eta > \frac{1+\gamma}{2}$.

*Table 2.* Simulation: Comparison of coverage and prediction interval width for five methods across three scenarios. The target coverage is 90%.

| Method | Scenario 1 | | Scenario 2 | | Scenario 3 | |
|---|---|---|---|---|---|---|
| | Coverage | Width | Coverage | Width | Coverage | Width |
| LSCP | $0.902_{\pm 0.005}$ | $\mathbf{1.08}_{\pm 0.05}$ | $0.907_{\pm 0.006}$ | $\mathbf{0.53}_{\pm 0.03}$ | $0.914_{\pm 0.006}$ | $\mathbf{0.75}_{\pm 0.05}$ |
| EnbPI | $0.878_{\pm 0.005}$ | $1.39_{\pm 0.05}$ | $0.883_{\pm 0.007}$ | $0.64_{\pm 0.03}$ | $0.883_{\pm 0.006}$ | $0.93_{\pm 0.04}$ |
| GSCP | $0.905_{\pm 0.009}$ | $1.55_{\pm 0.06}$ | $0.915_{\pm 0.004}$ | $0.73_{\pm 0.04}$ | $0.916_{\pm 0.005}$ | $1.1_{\pm 0.05}$ |
| SLSCP | $0.910_{\pm 0.006}$ | $1.42_{\pm 0.05}$ | $0.903_{\pm 0.004}$ | $0.67_{\pm 0.03}$ | $0.916_{\pm 0.005}$ | $0.95_{\pm 0.05}$ |
| LCP | $0.902_{\pm 0.008}$ | $1.54_{\pm 0.05}$ | $0.915_{\pm 0.006}$ | $0.58_{\pm 0.03}$ | $0.914_{\pm 0.005}$ | $1.06_{\pm 0.05}$ |
| SLCP | $0.902_{\pm 0.008}$ | $1.33_{\pm 0.05}$ | $0.900_{\pm 0.006}$ | $0.64_{\pm 0.03}$ | $0.894_{\pm 0.007}$ | $0.874_{\pm 0.05}$ |

*Table 3.* Real data analysis: Comparison of coverage and prediction interval width for five methods across five distinct datasets (New Mexico, Georgia, Farming, California Housing, and King County). The target coverage is 90%.

| Method | New Mexico | | Georgia | | Farming | | Cal. Housing | | King County | |
|---|---|---|---|---|---|---|---|---|---|---|
| | Cov. | Width | Cov. | Width | Cov. | Width | Cov. | Width | Cov. | Width |
| LSCP | 0.926 | **211.3** | 0.906 | **130.8** | 0.900 | **3,605.33** | 0.898 | **159,370.08** | 0.903 | **383,316.34** |
| EnbPI | 0.884 | 272.2 | 0.886 | 166.3 | 0.910 | 3,686.90 | 0.894 | 165,334.41 | 0.906 | 407,719.42 |
| GSCP | 0.901 | 295.9 | 0.892 | 167.3 | 0.910 | 3,687.25 | 0.895 | 196,959.02 | 0.903 | 419,219.85 |
| SLSCP | 0.898 | 276.8 | 0.896 | 167.8 | 0.900 | 3,694.17 | 0.893 | 164,571.93 | 0.910 | 401,277.29 |
| LCP | 0.901 | 283.9 | 0.893 | 166.7 | 0.900 | 3,702.04 | 0.899 | 197,504.70 | 0.904 | 420,069.28 |
| SLCP | 0.896 | 266.2 | 0.899 | 166.5 | 0.900 | 3,675.05 | 0.896 | 193,498.63 | 0.903 | 419,217.46 |

**Assumption 4.2** (Estimation quality). There exists a sequence $\{\delta_n\}_{n \geq 1}$ such that

$$\sum_{i=1}^{n} \|\hat{\varepsilon}(s_i) - \varepsilon(s_i)\|^2 \leq \frac{\delta_n^2}{M_n}, \quad (3)$$

$$\|\hat{\varepsilon}(s_{n+1}) - \varepsilon(s_{n+1})\| \leq \delta_n.$$

The assumption requires the average prediction error to be bounded by $\delta_n^2$, a weaker condition than in (Xu & Xie, 2021). Notably, our coverage gap result does not require $\delta_n$ to converge to zero, although this occurs in many settings. For example, research on neural network prediction error, such as (Barron, 1994), shows that under certain regularization conditions on $f$, $\delta_n = O(1/\sqrt{n})$.

**Assumption 4.3** (Stationary and spatial mixing). The random field $\varepsilon(s)$ is stationary and strongly mixing with coefficient $\alpha$, and the strong mixing coefficient can be bounded by $\alpha(a, b) \leq \alpha_1(a)g(b)$, where $\alpha_1$ is a nonincreasing function with $\lim_{a \to \infty} \alpha_1(a) = 0$. We assume $E_{d \sim g_n} \alpha_1(d)^2 \leq \frac{M}{n^2}$, where $g_n(d)$ is the distribution of the distance between two sample points $s_i$ and $s_j$ ($1 \leq i, j \leq n$).

Assumption 4.3 requires that the error process $\varepsilon(s)$ is spatially stationary and strongly mixing. Intuitively, the strong mixing condition captures a "decay of dependence": it implies that the statistical correlation between noise terms at distinct locations diminishes as the separation distance $d$ increases, vanishing asymptotically as $d \to \infty$. Crucially,

we emphasize that this assumption is imposed *solely* on the unobserved noise $\varepsilon(s)$, rather than on the distribution of the observations $\{(X(s), Y(s))\}$. This distinction is fundamental, as it allows the data $(X, Y)$ to exhibit complex, non-stationary patterns and arbitrary spatial structures, provided the residual errors remain stationary. Consequently, this condition is strictly weaker than the independent and identically distributed (i.i.d.) assumption relied upon in (Mao et al., 2024), serving as a natural multi-dimensional generalization of time-series mixing while retaining the i.i.d. setting as a special case.

**Assumption 4.4** (Lipschitz continuity). The true CDF $F_\varepsilon(y)$ of the noise $\varepsilon$ is assumed to be Lipschitz continuous with constant $L_{n+1}$.

### 4.4. Finite-sample marginal coverage

When we use the empirical quantile as the quantile estimator in Equation 1, LSCP reduces to the GSCP method. When spatial locations $s_1, \ldots, s_{n+1}$ are i.i.d., the spatial data $(X(s_i), Y(s_i))$ are exchangeable, and finite-sample marginal coverage can be proved in a standard way.

**Lemma 4.5** (Finite-sample marginal coverage (Mao et al., 2024)). *When $s_1, \ldots, s_{n+1}$ are i.i.d., then $(X(s_1), Y(s_1)), \ldots, (X(s_{n+1}), Y(s_{n+1}))$ is an exchangeable sequence and*

$$\mathbb{P}\Big(Y(s_{n+1}) \in \widehat{C}_n(X(s_{n+1}))\Big) \geq 1 - \alpha. \quad (4)$$

## 4.5. Conditional coverage

When the quantile is estimated by a weighted empirical distribution (e.g., via QRF), symmetry is lost, so exchangeability no longer implies finite-sample marginal coverage. Under the stationarity and spatial mixing assumptions stated above, we instead establish asymptotic conditional coverage guarantees for LSCP. Proofs and additional technical details are deferred to Appendix A.

Our main theorem is stated below. Using the tower law property, we can also establish the same result (Corollary A.5) for marginal coverage of LSCP.

**Theorem 4.6** (Conditional coverage). *Under Assumption 4.1–4.3, for any $\alpha \in (0,1)$ and sample size $T$, we have*

$$\left| \mathbb{P}\Big(Y(s_{n+1}) \in \widehat{C}_n(X(s_{n+1})) \mid X(s_{n+1}), s_{n+1}\Big) - (1-\alpha) \right|$$
$$\leq 4L_{n+1}\delta_n + 6M_n n^{\frac{1+\gamma}{2}} + (2 + 4\sqrt{M}g(b))(\log_2 n + 2)^2 n^{-\gamma}. \tag{5}$$

*When $n \to \infty$, we have*

$$\mathbb{P}\Big(Y(s_{n+1}) \in \widehat{C}_n(X(s_{n+1})) \mid X(s_{n+1}), s_{n+1}\Big) \to 1-\alpha. \tag{6}$$

From Inequality 5, we can see that the order of the coverage bound is controlled by $M_n n^{\frac{1+\gamma}{2}}$ and $n^{-\gamma}(\log_2 n + 2)^2$. The first term is equal to $n^{\frac{\gamma-1}{2}}$ in the special case of $M_n = \frac{1}{n}$, and the second term diminishes when $n$ is large enough because $n$ is of higher order than $\log_2 n$. As long as the estimation gap $\delta_n$ goes to zero as $n$ increases, the asymptotic conditional coverage can be inferred from the main theorem.

## 5. Experiments

In this section, we compare our proposed LSCP method with four baselines: EnbPI (Xu & Xie, 2021), which assigns equal weights to recent observations in time series; GSCP (Mao et al., 2024), which applies equal weights to all data points; SLSCP (Mao et al., 2024), which utilizes $k$-nearest neighbors and weights points based on spatial distance; and LCP (Guan, 2023), which weights all points according to feature similarity using a Gaussian kernel.

We randomly split the dataset into three subsets: $40\%$ for training, $40\%$ for calibration, and $20\%$ for testing. For each method and dataset, the number of neighbors and the Gaussian kernel bandwidth are selected using 5-fold cross-validation on the calibration set. The prediction model used in the experiments is the KNN regressor. More experimental details and choices of hyperparameters can be found in Appendix C. We also conduct a sensitivity analysis of the LSCP neighborhood size $k$ in Appendix C.6. The results indicate that LSCP is robust to $k$, with coverage varying only slightly across a wide range of values.

## 5.1. Synthetic data experiments

We evaluate the performance of LSCP against baseline methods across three simulated scenarios. Sampling locations $s$ are drawn uniformly from the unit grid $[0, 1]^2$. The latent signal $X(s)$ is modeled as a mean-zero stationary Gaussian process with a Matérn covariance function (variance $\sigma^2 = 1$, range $\phi = 0.1$, smoothness $\kappa = 0.7$). The response is generated according to the following three scenarios:

1. $Y(s) = X(s) + \epsilon(s)$,

2. $Y(s) = X(s)|\epsilon(s)|$,

3. $Y(s) = X(s) + \sin(\|s\|_2)\epsilon(s)$.

Here $\epsilon(s)$ denotes a noise process that is independent of the signal $X(s)$. These scenarios correspond to additive noise, signal-dependent noise, and spatially adaptive noise, respectively, and introduce nonlinearity and heteroskedasticity beyond the assumptions of our theoretical analysis. Despite this mismatch, the empirical results show that LSCP consistently outperforms baseline methods across all settings.

As shown in Table 2, LSCP achieves the nominal $90\%$ coverage in all scenarios while producing substantially narrower prediction intervals. The small standard deviations in both coverage and interval width indicate stable performance. Figure 3 visualizes the spatial distribution of interval widths for different methods. GSCP is omitted from these plots since it yields constant-width intervals across space. Interpreting color as uncertainty and comparing against the true residual heatmap, LSCP closely tracks sharp transitions and heteroskedastic regions, whereas EnbPI and SLSCP exhibit smoother, more homogenized patterns due to local averaging.

To further assess spatial reliability, we partition the grid into $10 \times 10$ cells and compute per-cell coverage and mean interval width, as shown in Figure 4. All methods except EnbPI achieve average coverage above the target $90\%$, with LSCP exhibiting the tightest dispersion and smallest widths overall. This indicates more effective use of data in low-uncertainty regions and appropriate widening in high-uncertainty areas. LCP behaves similarly to GSCP when the kernel bandwidth is large, as its weights approach uniformity over the calibration set.

## 5.2. Real-World Experiments

To demonstrate the robustness and versatility of LSCP, we evaluate it on a diverse set of real-world benchmarks spanning network connectivity, agriculture, and real estate. The evaluation includes: (1) mobile signal strength data from Georgia and New Mexico, which present extreme spatial imbalance and measurement noise; (2) a crop yield prediction dataset with limited sample size ($N = 500$), testing

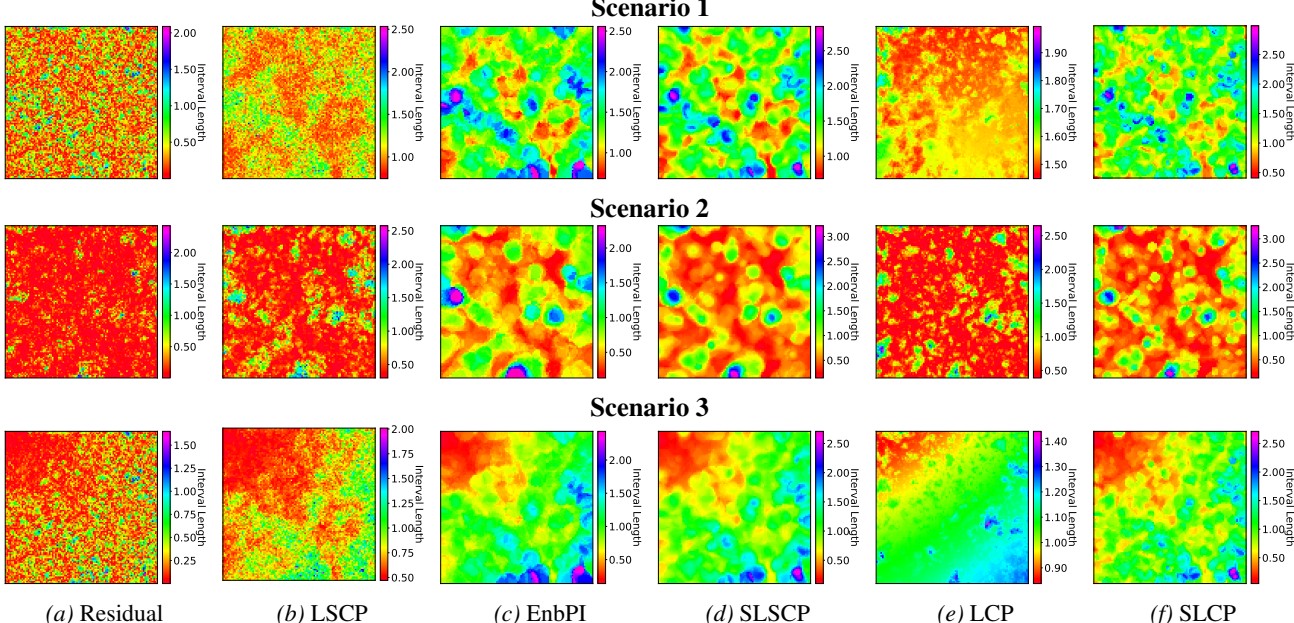

*Figure 3.* The heatmaps illustrate the width of the prediction intervals for each method across the three synthetic scenarios. The width heatmap of LSCP closely matches the true residual heatmap, demonstrating its ability to capture fine details accurately.

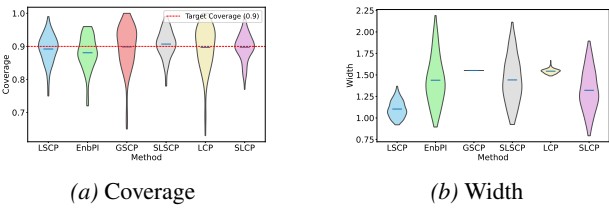

*(a)* Coverage  *(b)* Width

*Figure 4.* The violin plots on the left show the distribution of coverage across different areas for Scenario 1, while the plots on the right show the distribution of width.

adaptive weighting in LSCP mitigate these effects more effectively than the alternatives, yielding higher coverage and tighter intervals. Overall, LSCP consistently achieves high coverage with narrow prediction intervals. The violin plots further confirm its spatial stability and robustness to non-uniform data distributions, corroborating the synthetic results and validating its effectiveness in complex real-world settings.

## 6. Conclusion

We presented LSCP, a novel approach to uncertainty quantification for spatial data that addresses the limitations of fixed-kernel conformal methods. By learning data-adaptive weights from local residuals, LSCP avoids manual kernel specification and restrictive infill assumptions, making it effective for heterogeneous spatial environments. Our theoretical analysis establishes asymptotic conditional coverage guarantees for weighted conformal prediction under spatial mixing conditions. Extensive experiments on both synthetic and real-world datasets demonstrate that LSCP produces sharper prediction intervals while maintaining valid coverage, highlighting its robustness under complex spatial dependence. These properties make LSCP a promising foundation for future research. Moreover, the proposed theoretical framework naturally extends to spatio-temporal settings where analogous mixing conditions hold, offering a principled alternative to exchangeability-based approaches.

generalization under data scarcity; and (3) two large-scale housing datasets (California and King County), representing high-dimensional regression problems with substantial price variability. Further details of the datasets are provided in Appendix C.3.

As shown in Table 3, our proposed LSCP method outperforms competing methods by achieving substantially narrower prediction intervals while maintaining better coverage. Mirroring the synthetic study, we partition each state into $10 \times 10$ cells and compute per-cell coverage and mean interval width on the test set. The violin plots in Figure 4 highlight spatial consistency rather than global averages. Coverage is naturally higher in dense urban cells and lower in sparse rural regions. Notably, for New Mexico, the area-level coverage of all methods falls short of the nominal rate, whereas LSCP comes closest to the target. This shortfall is driven by severe data imbalance and sparsity in that state (Figure 1): some cells contain too few informative neighbors, which reduces local coverage regardless of method. Even under these challenging conditions, the locality and

## Impact Statement

This paper presents work whose goal is to advance the field of Machine Learning. There are many potential societal consequences of our work, none of which we feel must be specifically highlighted here

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

## A. Theoretical Details

### A.1. Derivation of Weighted Quantile

In our work, we employ the QRF from (Meinshausen, 2006) to construct the quantile estimator. Within this framework, the estimated CDF can be expressed as a weighted empirical quantile.

Consider a feature vector $\tilde{X}_i$ with support $\text{Supp}(\tilde{X}i) \subset \mathbb{B} \subset \mathbb{R}^p$. A regression tree $T(\theta)$ with parameter $\theta$ partitions $\mathbb{B}$ into $L$ disjoint rectangular subregions $R_l \subset \mathbb{B}$, each corresponding to a leaf node. By construction, these subregions cover $\mathbb{B}$, and every $x \in \mathbb{B}$ belongs to exactly one leaf, denoted $l(x, \theta)$. When $K$ trees are grown, each with its own parameter $\theta_k$, we define, for $x \in \mathbb{B}$ and observed features $\tilde{X}_1, \ldots, \tilde{X}_n$, the following weights:

$$k_\theta(l) := \#\{i \in \{1, \ldots, n\} : \tilde{X}_i \in R_{l(x,\theta)}\}, \tag{1}$$

$$\omega_i(x, \theta) := \frac{\mathbf{1}(\tilde{X}_i \in R_{l(x,\theta)})}{k_\theta(l)}, \tag{2}$$

$$\omega_i(x) := K^{-1} \sum_{k=1}^{K} \omega_i(x, \theta_k). \tag{3}$$

Here, (1) is the size of the leaf containing $x$, (2) assigns weight to the $i$-th observation according to its membership in this leaf normalized by leaf size, and (3) averages these weights across the $K$ trees. Using the aggregated weights, the estimated CDF is

$$\widehat{F}(y) := \sum_{i=1}^{n} \omega_i(x) \mathbf{1}(\tilde{Y}_i \le y). \tag{4}$$

Thus, $\widehat{F}(y)$ is a weighted empirical cumulative distribution function, where the weights are adaptively determined by the forest structure to reflect the proximity of each training point to the test input $x$ in feature space.

### A.2. Proofs

We first remind the readers of the notations introduced in the paper. In Appendix A, we have shown that our method is equivalent to using a weighted empirical CDF as the estimate of the true CDF, where the weights $\omega_i$ are learnt from data. The weighted empirical distribution for the true noise $\varepsilon$ is

$$\tilde{F}_{n+1}(y) = \sum_{i=1}^{n} \omega_i \mathbf{1}(\varepsilon(s_i) \le y). \tag{5}$$

Here $\omega_n$ is the normalized weight. Besides, we also define the weighted empirical distribution for the residual $\hat{\varepsilon}$ as

$$\widehat{F}_{n+1}(y) = \sum_{i=1}^{n} \omega_i \mathbf{1}(\hat{\varepsilon}(s_i) \le y). \tag{6}$$

We assume an additive true model which is commonly used in literature like (Xu & Xie, 2021):

$$Y(s) = f(X(s)) + \varepsilon(s). \tag{7}$$

Considering that the residual is $\hat{\varepsilon}(s) = Y(s) - \hat{f}(X(s))$, it follows

$$\varepsilon(s) - \hat{\varepsilon}(s) = \hat{f}(X(s)) - f(X(s)). \tag{8}$$

The following Lemma bounds the distance between the weighted empirical distribution for the residual and true error.

**Lemma A.1** (Distance between the empirical CDF of $\varepsilon$ and $\hat{\varepsilon}$). *Under Assumption 4.1 and 4.2,*

$$\sup_y |\widehat{F}_{n+1}(y) - \widetilde{F}_{n+1}(y)| \leq (L_{n+1} + 1)\delta_n + 2\sup_y |\widetilde{F}_{n+1}(y) - F_\varepsilon(y)|. \tag{9}$$

*Proof.* Using Assumption 4.2, we have that

$$\sum_{i=1}^n \omega_i |\varepsilon(s_i) - \hat{\varepsilon}(s_i)| \leq M_n \sum_{i=1}^n |\varepsilon(s_i) - \hat{\varepsilon}(s_i)| \leq \delta_n^2. \tag{10}$$

Let $S = \{i : |\varepsilon(s_i) - \hat{\varepsilon}(s_i)| \geq \delta_n\}$. Then

$$\delta_n \sum_{i \in S} \omega_i \leq \sum_{i=1}^n \omega_i |\varepsilon(s_i) - \hat{\varepsilon}(s_i)| \leq \delta_n^2. \tag{11}$$

So $\sum_{i \in S} \omega_i \leq \delta_n$. Then

$$\begin{aligned}
|\widehat{F}_{n+1}(y) - \widetilde{F}_{n+1}(y)| &\leq \sum_{i=1}^n \omega_i |\mathbf{1}\{\hat{\varepsilon}(s_i) \leq y\} - \mathbf{1}\{\varepsilon(s_i) \leq y\}| \\
&\leq \sum_{i \in S} \omega_i + \sum_{i \notin S} \omega_i |\mathbf{1}\{\hat{\varepsilon}(s_i) \leq y\} - \mathbf{1}\{\varepsilon(s_i) \leq y\}| \\
&\overset{(i)}{\leq} \sum_{i \in S} \omega_i + \sum_{i \notin S} \omega_i \mathbf{1}\{|\varepsilon(s_i) - y| \leq \delta_n\} \\
&\leq \sum_{i \in S} \omega_i + \sum_{i=1}^n \omega_i \mathbf{1}\{|\varepsilon(s_i) - y| \leq \delta_n\} \\
&\leq \delta_n + \mathbb{P}(|\varepsilon(s_{n+1}) - y| \leq \delta_n) + \\
&\quad \sup_y \left| \sum_{i=1}^n \omega_i \mathbf{1}\{|\varepsilon(s_i) - y| \leq \delta_n\} - \mathbb{P}(|\varepsilon(s_{n+1}) - y| \leq \delta_n) \right| \\
&= \delta_n + [F_\varepsilon(y + \delta_n) - F_\varepsilon(y - \delta_n)] + \sup_y \left| [\widetilde{F}_{n+1}(y + \delta_n) - \widetilde{F}_{n+1}(y - \delta_n)] \right. \\
&\quad \left. - [F_\varepsilon(y + \delta_n) - F_\varepsilon(y - \delta_n)] \right| \\
&\overset{(ii)}{\leq} (L_{n+1} + 1)\delta_n + 2\sup_y |\widetilde{F}_{n+1}(y) - F_\varepsilon(y)|,
\end{aligned}$$

where $(i)$ is because $|\mathbf{1}\{a \leq y\} - \mathbf{1}\{b \leq y\}| \leq \mathbf{1}\{|b - y| \leq |a - b|\}$ for $a, b \in \mathbb{R}$ and $(ii)$ is because the Lipschitz continuity of $F_\varepsilon(y)$. $\qquad\square$

**Lemma A.2.** *Under assumption 4.1-4.3, we have*

$$E\left(\sup_y |\tilde{F}_{n+1}(y) - F_\varepsilon(y)|^2\right) \leq \frac{(2 + \log_2 n)^2}{n}(1 + 2\sqrt{M}g(b)). \tag{12}$$

*Proof.* Define $Z_{n+1}(y) = \tilde{F}_{n+1}(y) - F_\varepsilon(y)$. Besides, we also define $Z_{n+1}(A) = \sum_{i=1}^n \omega_i \mathbf{1}(X(s_i) \in A) - F_\varepsilon(\varepsilon \in A)$, where $A$ can be any region. Let $N$ be some positive integer to be chosen later. We first represent the CDF $F_\varepsilon(x)$ in base 2:

$$F_\varepsilon(x) = \sum_{i=1}^N b_i(x)2^{-i} + r_N(x), \tag{13}$$

where $r_N(x) \in [0, 2^{-N})$ and $b_i = 0$ or 1.

For $l \in \{1, 2, \cdots, N\}$, define

$$B_l(x) = \sum_{i=1}^{l} b_i(x) 2^{-i}. \tag{14}$$

We can define the points $x_i$ where $F_\varepsilon(x_i) = B_i(x)$. We have that

$$F_\varepsilon(x) - F_\varepsilon(x_i) = \sum_{i=l+1}^{N} a_i(x) 2^{-i} + r_N(x) \le 2^{-l}. \tag{15}$$

As a result, we can partition $Z_{n+1}(x)$ into the following sum:

$$
\begin{aligned}
Z_{n+1}(F_\varepsilon^{-1}(x)) = Z_{n+1}(F_\varepsilon^{-1}(B_1(x))) + \sum_{i=1}^{N-1} (Z_{n+1}(F_\varepsilon^{-1}(B_{i+1}(x))) - Z_{n+1}(F_\varepsilon^{-1}(B_i(x)))) \\
+ (Z_{n+1}(F_\varepsilon^{-1}(y)) - Z_{n+1}(F_\varepsilon^{-1}(B_N(x)))).
\end{aligned}
\tag{16}
$$

In order to bound $Z_{n+1}(y)$, we can bound each individual term instead. Since $B_{i+1}(x) - B_i(x) = b_{i+1}(x) 2^{-(i+1)} \le 2^{-(i+1)}$, we know the interval $[B_i(x), B_{i+1}(x)]$ either has zero length, or it is equal to one region in the set $\{[(j-1)2^{-(i+1)}, j2^{-(i+1)}], 1 \le j \le 2^{i+1}\}$. As a result, we have

$$
\begin{aligned}
|Z_{n+1}(F_\varepsilon^{-1}(B_{i+1}(x))) - Z_{n+1}(F_\varepsilon^{-1}(B_i(x)))| \le \\
\sup_{j \in [1, 2^{i+1}]} |Z_{n+1}(F_\varepsilon^{-1}(j2^{-(i+1)})) - Z_{n+1}(F_\varepsilon^{-1}((j-1)2^{-(i+1)}))|
\end{aligned}
\tag{17}
$$

Let

$$\delta_i = \sup_{j \in [1, 2^{i+1}]} |Z_{n+1}([F_\varepsilon^{-1}(j2^{-(i+1)}), F_\varepsilon^{-1}((j-1)2^{-(i+1)})])|,$$

and

$$\delta_{xN} = \sup_x |Z_{n+1}([F_\varepsilon^{-1}(B_N(x)), x])|.$$

It follows that

$$|Z_{n+1}(F_\varepsilon^{-1}(y))| \le \sum_{i=1}^{N} \delta_i + \delta_{xN}. \tag{18}$$

By the triangle inequality,

$$(E(\sup_{y \in [0,1]} |Z_{n+1}(F_\varepsilon^{-1}(y))|^2))^{1/2} \le \sum_{i=1}^{N} (E\delta_i^2)^{1/2} + (E\delta_{xN}^2)^{1/2}. \tag{19}$$

Then we need to bound $\|\delta_i\|_2$ and $\|\delta_{xN}\|_2$ separately. Since $\delta_i$ is computing the supremum over a set, it can be bounded by the sum over the set,

$$\delta_i^2 \le \sum_{j=1}^{2^i} (Z_{n+1}(F_\varepsilon^{-1}(j2^{-(i+1)})) - Z_{n+1}(F_\varepsilon^{-1}((j-1)2^{-(i+1)})))^2. \tag{20}$$

Taking expectation, we have

$$
\begin{aligned}
E\delta_i^2 &\le \sum_{j=1}^{2^i} E(Z_{n+1}(F_\varepsilon^{-1}(j2^{-(i+1)})) - Z_{n+1}(F_\varepsilon^{-1}((j-1)2^{-(i+1)})))^2 \\
&= \sum_{j=1}^{2^i} \text{Var}(Z_{n+1}([F_\varepsilon^{-1}((j-1)2^{-(i+1)}), F_\varepsilon^{-1}(j2^{-(i+1)})])).
\end{aligned}
\tag{21}
$$

Let $(\epsilon_i)_{i>0}$ be a sequence of independent and symmetric random variables in $\{-1, 1\}$. For any finite partition $A_1, \cdots, A_k$ of $\mathbb{R}$,

$$\sum_{j=1}^{k} \text{Var}\, Z_{n+1}(A_j) = E(Z_{n+1}^2(\sum_{j=1}^{k} \epsilon_i \mathbf{1}_{A_i}))$$

$$\overset{(i)}{\leq} M_n^2(n + 2E \sum_{1 \leq i < j \leq n} \alpha_{ij}), \tag{22}$$

where $\alpha_{ij} = \alpha(\sigma(\varepsilon(s_i)), \sigma(\varepsilon(s_j)))$ is the alpha-mixing coefficient, $M_n = \max_{1 \leq i \leq n} \omega_i$ and $(i)$ is because of Lemma 1.1 in (Rio et al., 2017).

Because of Assumption 4.3, we have

$$E \sum_{1 \leq i < j \leq n} \alpha_{ij} \leq E \sum_{1 \leq i < j \leq n} \alpha_1(|s_i - s_j|)g(b)$$

$$\leq n \sqrt{E \sum_{1 \leq i < j \leq n} \alpha_1^2(|s_i - s_j|)g(b)} \tag{23}$$

$$\leq n^2 \sqrt{E_{d \sim g_n} \alpha_1^2(d)g(b)}$$

$$\leq n\sqrt{M}g(b),$$

where $g_n$ is the distribution of the distance between $s_i$ and $s_j$ for any $i, j \in \{1, \cdots, n\}$.

Because $[F_\varepsilon^{-1}((j-1)2^{-(i+1)}), F_\varepsilon^{-1}(j2^{-(i+1)})]$ for $j = 1, \cdots, 2^{i+1}$ is a partition of $\mathbb{R}$, from 22,

$$E\delta_i^2 \leq \sum_{j=1}^{2^i} \text{Var}(Z_{n+1}([F_\varepsilon^{-1}((j-1)2^{-(i+1)}), F_\varepsilon^{-1}(j2^{-(i+1)})]))$$

$$\leq nM_n^2(1 + 2\sqrt{M}g(b)). \tag{24}$$

For the other term $\delta_{xN}$, we know $x = F_\varepsilon^{-1}(F_\varepsilon(x)) \leq F_\varepsilon^{-1}(B_N(x) + r_N(x))$. We have

$$Z_{n+1}([F_\varepsilon^{-1}(B_N(x)), x]) = \tilde{F}_{n+1}([F_\varepsilon^{-1}(B_N(x)), x]) - F_\varepsilon([F_\varepsilon^{-1}(B_N(x)), x])$$

$$\geq -F_\varepsilon([F_\varepsilon^{-1}(B_N(x)), x]) \tag{25}$$

$$= B_N(x) - x \geq -2^{-N}.$$

On the other hand,

$$Z_{n+1}([F_\varepsilon^{-1}(B_N(x)), x]) = Z_{n+1}([F_\varepsilon^{-1}(B_N(x)), B_N(x)) + 2^{-N}]) - Z_{n+1}([F_\varepsilon^{-1}(x, B_N(x)) + 2^{-N}])$$

$$\leq Z_{n+1}([F_\varepsilon^{-1}(B_N(x)), B_N(x)) + 2^{-N}]) + 2^{-N}. \tag{26}$$

As a result, we have

$$\delta_{xN} \leq \delta_N + 2^{-N}. \tag{27}$$

To sum up, we prove that

$$(E(\sup_{y \in [0,1]} |Z_{n+1}(F_\varepsilon^{-1}(y))|^2))^{\frac{1}{2}} \leq n^{\frac{1}{2}} M_n(N + 1 + 2^{-N})(1 + 2\sqrt{M}g(b))^{\frac{1}{2}}. \tag{28}$$

Let $N = \log_2 n$, we have

$$E(\sup_{y \in [0,1]} |Z_{n+1}(F_\varepsilon^{-1}(y))|^2) \leq M_n^2 n(2 + \log_2 n)^2(1 + 2\sqrt{M}g(b)). \tag{29}$$

$\square$

**Lemma A.3** (Convergence of empirical CDF of $\varepsilon$). *Under Assumptions 4.1-4.3, with probability higher than* $1 - (1 + 2\sqrt{M}g(b))(\log_2 n + 2)^2 n^{-\gamma}$,

$$\sup_y \left| \widetilde{F}_{n+1}(y) - F_\epsilon(y) \right| \leq M_n n^{\frac{1+\gamma}{2}}. \tag{30}$$

*Proof.* We have that

$$\tilde{F}_{n+1}(y) - F_\epsilon(y) = \sum_{i=1}^n \omega_i \mathbf{1}(\epsilon(s_i) \leq y) - F_\epsilon(y) \tag{31}$$

$$= \sum_{i=1}^n \omega_i (\mathbf{1}(\epsilon(s_i) \leq y) - F_\epsilon(y)).$$

Let $Z(s_i) = \mathbf{1}(\epsilon(s_i) \leq y) - F_\epsilon(y)$, we know

$$EZ(s_i) = 0, \tag{32}$$

and $Z(s)$ is stationary. From Lemma A.2, using Markov inequality, we have that for any $k$,

$$\mathbb{P}(\sup_y |\tilde{F}_{n+1}(y) - F_\varepsilon(y)| \geq k) \leq \frac{E(\sup_y |\tilde{F}_{n+1}(y) - F_\varepsilon(y)|^2)}{k^2} \tag{33}$$

$$= \frac{M_n^2 n (2 + \log_2 n)^2 (1 + 2\sqrt{M}g(b))}{k^2}.$$

Let $k = M_n n^{\frac{1+\gamma}{2}}$, we have

$$\mathbb{P}\left( \sup_y |\tilde{F}_{n+1}(y) - F_\varepsilon(y)| \geq M_n n^{\frac{1+\gamma}{2}} \right) \leq (2 + \log_2 n)^2 (1 + 2\sqrt{M}g(b)) n^{-\gamma}. \tag{34}$$

$\square$

**Theorem A.4.** *Under Assumption 4.1-4.3, for any $\alpha \in (0,1)$ and sample size $T$, we have*

$$\left| \mathbb{P}\left( Y(s_{n+1}) \in \widehat{C}_n \left( X(s_{n+1}) \right) \mid X(s_{n+1}), s_{n+1} \right) - (1 - \alpha) \right|$$
$$\leq 4L_{n+1}\delta_n + 6M_n n^{\frac{1+\gamma}{2}} + (2 + 4\sqrt{M}g(b))(\log_2 n + 2)^2 n^{-\gamma}. \tag{35}$$

*Proof.* For simplicity, we use $X_i = X(s_i)$, $Y_i = Y(s_i)$, $\varepsilon_i = \varepsilon(s_i)$, $\hat{\varepsilon}_i = \hat{\varepsilon}(s_i)$ and $\omega_{ni} = \omega_i$. For any $\beta \in [0,1]$,

$$\left| \mathbb{P}\left( Y_{n+1} \in \widehat{C}_n \left( X_{n+1} \right) \middle| X_{n+1}, s_{n+1} \right) - (1 - \alpha) \right|$$

$$= \left| \mathbb{P}\left( \hat{\varepsilon}_{n+1} \in [\widehat{Q}_\beta(X_{n+1}), \widehat{Q}_{1-\alpha+\beta}(X_{n+1})] \middle| X_{n+1}, s_{n+1} \right) - (1 - \alpha) \right|$$

$$= \left| \mathbb{P}\left( \hat{\varepsilon}_{n+1} \in [\widehat{Q}_\beta(X_{n+1}), \widehat{Q}_{1-\alpha+\beta}(X_{n+1})] \middle| X_{n+1} \right) - (1 - \alpha) \right|$$

$$= \left| \mathbb{P}\left( \beta \leq \sum_{i=1}^n \omega_{ni} \mathbf{1}(\hat{\varepsilon}_i \leq \hat{\varepsilon}_{n+1}) \leq 1 - \alpha + \beta \middle| X_{n+1} \right) - (1 - \alpha) \right|$$

$$= \left| \mathbb{P}\left( \beta \leq \widehat{F}_{n+1}(\hat{\varepsilon}_{n+1}) \leq 1 - \alpha + \beta \middle| X_{n+1} \right) - \mathbb{P}\left( \beta \leq F_\varepsilon(\varepsilon_{n+1}) \leq 1 - \alpha + \beta \right) \right|$$

$$= \left| \mathbb{P}\left( \beta \leq \widehat{F}_{n+1}(\hat{\varepsilon}_{n+1}) \leq 1 - \alpha + \beta \middle| X_{n+1} \right) - \mathbb{P}\left( \beta \leq F_\varepsilon(\varepsilon_{n+1}) \leq 1 - \alpha + \beta \middle| X_{n+1} \right) \right|$$

$$\leq \mathbb{E}\left( \left| \mathbf{1}\{\beta \leq \widehat{F}_{n+1}(\hat{\varepsilon}_{n+1}) \leq 1 - \alpha + \beta\} - \mathbf{1}\{\beta \leq F_\varepsilon(\varepsilon_{n+1}) \leq 1 - \alpha + \beta\} \right| \middle| X_{n+1} \right)$$

$$\overset{(i)}{\leq} \mathbb{E}\left( \left| \mathbf{1}\{\beta \leq \widehat{F}_{n+1}(\hat{\varepsilon}_{n+1})\} - \mathbf{1}\{\beta \leq F_\varepsilon(\varepsilon_{n+1})\} \right| \right.$$

$$+ \left| \mathbf{1}\{\widehat{F}_{n+1}(\hat{\varepsilon}_{n+1}) \leq 1 - \alpha + \beta\} - \mathbf{1}\{F_{\varepsilon}(\varepsilon_{n+1}) \leq 1 - \alpha + \beta\} \right| \Bigg| X_{n+1} \Bigg)$$

$$\overset{(ii)}{\leq} \mathbb{P}\left( \left| F_{\varepsilon}(\varepsilon_{n+1}) - \beta \right| \leq \left| F_{\varepsilon}(\varepsilon_{n+1}) - \widehat{F}_{n+1}(\hat{\varepsilon}_{n+1}) \right| \Bigg| X_{n+1} \right)$$

$$+ \mathbb{P}\left( \left| F_{\varepsilon}(\varepsilon_{n+1}) - (1 - \alpha + \beta) \right| \leq \left| F_{\varepsilon}(\varepsilon_{n+1}) - \widehat{F}_{n+1}(\hat{\varepsilon}_{n+1}) \right| \Bigg| X_{n+1} \right),$$

where Inequality $(i)$ follows since for any constants $a, b$ and univariates $x, y$, $|\, \mathbf{1}\{a \leq x \leq b\} - \mathbf{1}\{a \leq y \leq b\}| \leq |\mathbf{1}\{a \leq x\} - \mathbf{1}\{a \leq y\}| + |\mathbf{1}\{x \leq b\} - \mathbf{1}\{y \leq b\}|$. On the other hand, Inequality $(ii)$ is a result of $|\mathbf{1}\{a \leq x\} - \mathbf{1}\{b \leq x\}| \leq \mathbf{1}\{|b - x| \leq |a - b|\}$.

Using Lemma A.2,

$$\mathbb{P}\left( \left| F_{\varepsilon}(\varepsilon_{n+1}) - \beta \right| \leq \left| F_{\varepsilon}(\varepsilon_{n+1}) - \widehat{F}_{n+1}(\hat{\varepsilon}_{n+1}) \right| \Bigg| X_{n+1} \right)$$

$$\leq \mathbb{P}\left( \left| F_{\varepsilon}(\varepsilon_{n+1}) - \beta \right| \leq \left| F_{\varepsilon}(\varepsilon_{n+1}) - \widehat{F}_{n+1}(\hat{\varepsilon}_{n+1}) \right|, \sup_y \left| F_{\varepsilon}(y) - \widehat{F}_{n+1}(y) \right| \leq M_n n^{\frac{1+\gamma}{2}} \Bigg| X_{n+1} \right)$$

$$+ \mathbb{P}\left( \sup_y \left| F_{\varepsilon}(y) - \widehat{F}_{n+1}(y) \right| \geq M_n n^{\frac{1+\gamma}{2}} \Bigg| X_{n+1} \right)$$

$$\leq \mathbb{P}\left( \left| F_{\varepsilon}(\varepsilon_{n+1}) - \beta \right| \leq \left| F_{\varepsilon}(\varepsilon_{n+1}) - \widehat{F}_{n+1}(\hat{\varepsilon}_{n+1}) \right| \Bigg| \sup_y \left| F_{\varepsilon}(y) - \widehat{F}_{n+1}(y) \right| \leq M_n n^{\frac{1+\gamma}{2}}, X_{n+1} \right)$$

$$+ \mathbb{P}\left( \sup_y \left| F_{\varepsilon}(y) - \widehat{F}_{n+1}(y) \right| \geq M_n n^{\frac{1+\gamma}{2}} \right)$$

$$\leq \mathbb{P}\left( \left| F_{\varepsilon}(\varepsilon_{n+1}) - \beta \right| \leq |F_{\varepsilon}(\varepsilon_{n+1}) - F_{\varepsilon}(\hat{\varepsilon}_{n+1})| + (L_{n+1} + 1)\delta_n + 3M_n n^{\frac{1+\gamma}{2}} \Bigg| X_{n+1} \right)$$

$$+ (1 + 2\sqrt{M}g(b))(\log_2 n + 2)^2 n^{-\gamma}$$

$$\leq \mathbb{P}\left( \left| F_{\varepsilon}(\varepsilon_{n+1}) - \beta \right| \leq L_{n+1}|\varepsilon_{n+1} - \hat{\varepsilon}_{n+1}| + (L_{n+1} + 1)\delta_n + 3M_n n^{\frac{1+\gamma}{2}} \Bigg| X_{n+1} \right)$$

$$+ (1 + 2\sqrt{M}g(b))(\log_2 n + 2)^2 n^{-\gamma}$$

$$\leq 2L_{n+1}\delta_n + 3M_n n^{\frac{1+\gamma}{2}} + (1 + 2\sqrt{M}g(b))(\log_2 n + 2)^2 n^{-\gamma}.$$

The inequality above also holds for $\mathbb{P}\left( |F_{\varepsilon}(\varepsilon_{n+1}) - \beta| \leq |F_{\varepsilon}(\varepsilon_{n+1}) - \widehat{F}_{n+1}(\hat{\varepsilon}_{n+1})| \Big| X_{n+1} \right)$, and we can conclude that

$$\left| \mathbb{P}\left( Y_{n+1} \in \widehat{C}_n\left(X_{n+1}\right) \mid X_{n+1}, s_{n+1} \right) - (1 - \alpha) \right|$$
$$\leq 4L_{n+1}\delta_n + 6M_n n^{\frac{1+\gamma}{2}} + (2 + 4\sqrt{M}g(b))(\log_2 n + 2)^2 n^{-\gamma}. \tag{36}$$

$\square$

**Corollary A.5** (Marginal Coverage). *Under Assumption 4.1-4.3, for any $\alpha \in (0, 1)$ and sample size $T$, we have*

$$\left| \mathbb{P}\left( Y(s_{n+1}) \in \widehat{C}_n\left(X(s_{n+1})\right) \right) - (1 - \alpha) \right|$$
$$\leq 4L_{n+1}\delta_n + 6M_n n^{\frac{1+\gamma}{2}} + (2 + 4\sqrt{M}g(b))(\log_2 n + 2)^2 n^{-\gamma}. \tag{37}$$

# B. Comparison with other methods

**LSCP vs. GSCP**   Global Spatial Conformal Prediction (GSCP), introduced in (Mao et al., 2024), applies equal weighting to all non-conformity scores across the calibration dataset, and the estimated quantile at any point is given by

$$\widehat{Q}_n(p) = \inf\{e \in \mathbb{R} : \frac{1}{n}\sum_{i=1}^{n} \mathbf{1}\{\hat{\varepsilon}(s_i) \leq e\} \leq p\}.$$

This approach can be viewed as an extension of split conformal prediction and its key strength lies in the coverage guarantee obtained under minimal assumptions. In particular, when spatial locations are sampled i.i.d. from a common distribution, the data are exchangeable, as permuting the order does not alter the joint probability law. In this case, GSCP naturally ensures valid marginal coverage.

In practice, however, GSCP is often too conservative because real-world data distributions vary across locations. The method enforces uniform weights on all the training data, and captures local adaptivity only through a user-specified variance estimate $\hat{\sigma}$ in the non-conformity score. As a result, GSCP intervals are typically achieving coverage by being unnecessarily wide in some regions. A natural remedy is to localize the procedure by estimating empirical quantiles within neighborhoods, but doing so breaks the spatial exchangeability, and thus the theoretical guarantee no longer holds.

**LSCP vs. SLSCP**   Smoothed Local Spatial Conformal Prediction (SLSCP) is another method from (Mao et al., 2024) that improves GSCP by using only nearby data to construct prediction intervals, with the weighted empirical quantile estimated locally as:

$$\widehat{Q}_n(p) = \inf\{e \in \mathbb{R} : \sum_{i \in N(s_{n+1})} \omega_i \mathbf{1}\{\hat{\varepsilon}(s_i) \leq e\} \leq p\}.$$

SLSCP assigns weights $\omega_i \propto k(\|s_i - s_{n+1}\|)$, relying solely on spatial distance. In contrast, our method learns data-adaptive $\omega_i$ by training quantile regression with features $\tilde{X}(s)$, capturing richer information beyond distance. As shown in Section 5, this makes our approach more adaptive and significantly improves performance over SLSCP.

Our theoretical framework also differs fundamentally from theirs. SLSCP relies on infill sampling assumption, requiring data points to become arbitrarily dense around $s_{n+1}$, and assumes a spatially continuous process with locally i.i.d. noise. These conditions yield local asymptotic exchangeability, which lead to coverage guarantee. By contrast, our approach assumes only a stationary, spatially mixing error process, our method avoids the need for infinitely close neighbors and generalizes more broadly. Moreover, this framework naturally extends to spatio-temporal settings, making LSCP applicable in real-world scenarios where SLSCP fails—particularly for time series data, which violate the infill assumption.

**LSCP vs. LCP**   Localized Conformal Prediction (LCP), introduced in (Guan, 2023), provides a general framework for localized conformal prediction rather than the spatial setting only. The method combines GSCP and SLSCP in quantile estimation:

$$\widehat{Q}_n(p) = \inf\{e \in \mathbb{R} : \sum_{i=1}^{n} \omega_i \mathbf{1}\{\hat{\varepsilon}(s_i) \leq e\} \leq p\}.$$

Similar to GSCP, LCP uses all calibration data for prediction, but like SLSCP, it applies different weights to each data point, which are defined as $\omega_i \propto k(X(s_i), X(s_{n+1}))$, where $k$ is a user-specified kernel function. While this kernel-based design provides more flexibility than purely location-based weights in SLSCP, it has two key limitations: (i) it only encodes pairwise similarity between features, and (ii) the kernel must be hand-specified by the user, which makes performance highly sensitive to this choice.

Our proposed LSCP removes these limitations by directly learning data-adaptive weights. Instead of relying on a user-specified kernel, LSCP uses quantile regression on neighborhood residuals to infer weights automatically from the data. This not only captures richer local dependencies that fixed kernels cannot, but also makes the method far easier to apply in practice, since it eliminates the need to tune or select a kernel function. As a result, LSCP is both more expressive and more user-friendly than LCP, while achieving stronger empirical performance.

Another limitation of LCP lies in its theoretical assumptions: it requires the data $(X_i, Y_i)_{i=1}^{n}$ to be i.i.d. to guarantee finite-sample marginal coverage. This assumption is stronger than those of other conformal methods mentioned, and it restricts the generality and applicability of the results.

# C. Experimental Details

## C.1. Baseline Methods

In our experiments, we compare LSCP against five baselines spanning time series and spatial conformal prediction:

1. **EnbPI** (Xu & Xie, 2021): a general framework for time series prediction intervals that fits leave-one-out regressors and uses their residuals as nonconformity scores. Quantiles are computed from the empirical (uniformly weighted) residual distribution from a past window.

2. **GSCP** (Mao et al., 2024): a spatial CP method that extends split conformal to spatial settings by using a global empirical quantile with uniform weights over all calibration points.

3. **SLSCP** (Mao et al., 2024): a localized spatial CP method that estimates a weighted quantile using a distance-based kernel, thereby adapting to local spatial structure.

4. **LCP** (Guan, 2023): a general localized CP framework that assigns weights to all calibration points via a user-specified kernel on feature similarity, aiming to capture feature-level dependence.

5. **SLCP** (Han et al., 2022): a split localized conformal prediction method that partitions the feature space (e.g., via a learned or predefined grouping) and computes split-conformal quantiles within each local region.

## C.2. Data Simulation

In order to compare the performance of different methods, we simulate several different scenarios that go beyond the assumptions made in the paper.

We uniformly sample the spatial locations $s$ from the unit grid $[0, 1] \times [0, 1]$. For each scenario, we sample $12000$ data points, where $40\%$ are used as training data, $40\%$ as calibration data and the rest $20\%$ as test data. Let $X(s)$ be a mean-zero stationary Gaussian process with Matérn covariance (variance $\sigma^2 = 1$, range $\phi = 0.1$, smoothness $\kappa = 0.7$). We use the three regimes that have been considered in spatial CP literature:
1. Stationary, homoskedastic spatial data:

$$Y(s) = X(s) + \varepsilon(s).$$

This scenario establishes a reference case close to standard assumptions (stationary signal with additive, location-invariant noise), ensuring methods behave sensibly when global structure suffices.

2. Heteroskedastic spatial data:

$$Y(s) = X(s)|\varepsilon(s)|.$$

This scenario introduces heteroskedasticity tied to the latent field (variance depends on $|X(s)|$) and non-Gaussian noise, stresses robustness to model misspecification and the ability to adapt via local residual information.

3. Heteroskedastic and Non-stationary spatial data:

$$Y(s) = X(s) + \sin(\|s\|_2)\varepsilon(s).$$

This scenario creates spatially varying uncertainty (larger near peaks of $\sin(\|s\|_2)$, smaller near zeros), violates stationarity and highlights the benefit of localization that adapts to spatial heterogeneity.

## C.3. Real-data Description

**Mobile Signal Strength**   The dataset utilized in our study is sourced from the open datasets provided by Ookla. The data gathered by Ookla from 2019 to 2023 encapsulates the performance metrics of mobile internet connections for a multitude of users worldwide. Key variables in this dataset include geographical coordinates (longitude and latitude), mean download speed (MB/s), mean upload speed (MB/s), count of tests conducted in each area (aggregated for user privacy into $600m^2$ grid blocks), the number of distinct devices utilized for testing, and a comprehensive speed score assessing the connection speed. In our experiment, we utilize mobile connection data from the states of Georgia in the United States Southeast and New Mexico in the United States Southwest. In our experiments, we use mobile connection data from Georgia and New Mexico (USA) and aim to predict the speed score at new spatial locations. The New Mexico dataset contains $24,983$ observations, while the Georgia dataset includes $28,587$ data points.

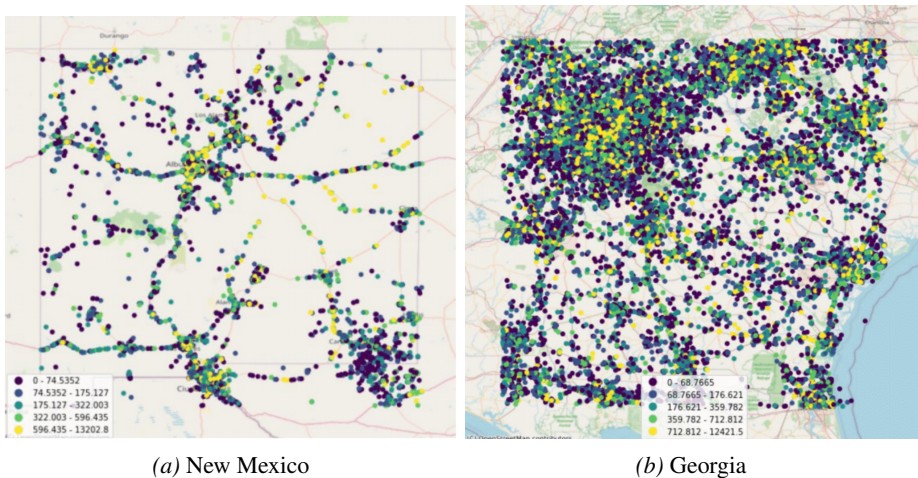

*(a)* New Mexico    *(b)* Georgia

*Figure 1.* Mobile signal score from Ookla dataset. The plots show the distribution of data accross the state.

**Crop Yield Prediction.**    We then analyze a farming dataset that captures environmental and operational variables affecting crop yield across 500 farms situated in diverse geographical regions, including India, the USA, and Africa.

**Real Estate Price Prediction.**    We further assess the method on two large-scale housing datasets. The first is the California Housing dataset, comprising $20,640$ census data points. The second is the King County House Sales dataset, containing $21,613$ transactions.

### C.4. Hyperparameter Choice

In our experiments we fix the base predictor to a k-nearest-neighbors regressor with $k = 5$, where $k$ is chosen by 5-fold cross validation. This keeps the mean model simple and transparent while letting us focus on how different uncertainty wrappers behave. In principle, any reasonable prediction model could replace KNN without changing the conformal machinery.

For locality-based methods (LSCP, EnbPI and SLSCP), we use neighborhoods of 50 points. Empirically, coverage and width are fairly insensitive to this choice once the neighborhood is large enough to yield stable empirical quantiles but not so large that it washes out spatial heterogeneity.

For the kernel-smoothed methods (SLSCP and LCP), We tune it over $[0, 1]$ via five-fold cross-validation, selecting the smallest bandwidth that achieves nominal coverage on held-out folds while minimizing interval width. If no choice achieves the nominal coverage, then we choose the one that generates the closest coverage to it. The choice is $0.03$ for SLSCP and $0.5$ for LCP.

Finally, for quantile model in LSCP we use a Quantile Random Forest with 50 trees and maximum depth 10. Deeper trees tend to produce slightly tighter intervals by capturing finer structure, but at the cost of increased computation and a higher risk of overfitting in small local neighborhoods.

### C.5. Computational Complexity

A potential concern with localized methods is the computational cost associated with fitting models on large spatial datasets. It is important to clarify that LSCP does *not* require training a separate model for each test location. Instead, the method maintains scalability through a two-stage design: (i) training a **single, global** Quantile Random Forest (QRF) on the residual-feature pairs, and (ii) performing lightweight, local information retrieval at inference time.

**Training Complexity.**    The training phase consists of fitting one QRF model on the calibration residuals $\{(\tilde{X}(s_i), \tilde{Y}(s_i))\}_{i=1}^{n}$. The computational complexity is $\mathcal{O}(T \cdot n \log n)$, where $n$ is the sample size and $T$ is the number of trees. This cost is identical to fitting a standard Random Forest and scales efficiently to datasets with millions of points using parallelized implementations.

**Inference Complexity.** For a new test location $s_{n+1}$, the computational cost involves two steps:

1. **Neighbor Search:** Retrieving the $k$-nearest spatial neighbors. Using spatial indexing structures such as Ball Trees or $k$-d trees, this query can be performed in $\mathcal{O}(k \log n)$ time.

2. **Model Evaluation:** A single forward pass of the pre-trained QRF. Since the input dimension is determined by the neighborhood size $k$ (which is fixed and small), this step is $\mathcal{O}(T \cdot d_{\max})$, where $d_{\max}$ is the maximum depth of the trees.

Because the input dimension to the QRF depends only on $k$ and not on the total dataset size $n$, the inference cost remains low even as the spatial domain grows. Consequently, the total complexity of LSCP is comparable to standard global conformal prediction pipelines, making it tractable for massive spatial datasets.

### C.6. Sensitivity Analysis

We investigate the sensitivity of the LSCP method to the choice of the neighborhood size $k$, which determines the locality of the weighted conformal prediction. Table 1 reports the empirical coverage and average interval width across a wide range of values, $k \in \{5, \dots, 150\}$, for both the New Mexico and Georgia datasets. The results demonstrate that the proposed method is highly robust to the selection of $k$. The empirical coverage remains stable and consistently above the nominal level (ranging from 0.923 to 0.926 for NM and 0.905 to 0.909 for GA). Similarly, the average interval width exhibits only minor fluctuations, increasing slightly as $k$ grows large but maintaining efficiency compared to other baseline methods. This stability suggests that LSCP does not require precise hyperparameter tuning to achieve valid and efficient inference.

| $k$ | New Mexico | | Georgia | |
|---|---|---|---|---|
| | Coverage | Width | Coverage | Width |
| 5 | 0.923 | 208.40 | 0.908 | 129.29 |
| 8 | 0.923 | 211.63 | 0.908 | 130.17 |
| 10 | 0.925 | 212.56 | 0.908 | 130.95 |
| 15 | 0.924 | 213.22 | 0.907 | 132.67 |
| 20 | 0.926 | 213.41 | 0.905 | 131.53 |
| 30 | 0.924 | 213.33 | 0.909 | 132.49 |
| 50 | 0.925 | 216.25 | 0.909 | 132.86 |
| 75 | 0.924 | 216.36 | 0.909 | 133.05 |
| 100 | 0.924 | 217.96 | 0.906 | 133.13 |
| 150 | 0.923 | 219.94 | 0.907 | 133.97 |

*Table 1.* Sensitivity of LSCP coverage and average interval width to the neighborhood size $k$ on the NM and GA datasets.

### C.7. Additional experiment results

In this section, we present residual heatmaps and violin plots to further visualize our experimental results. The residual heatmaps in Figure 3 demonstrate that LSCP accurately tracks sharp transitions and heteroskedastic pockets across all datasets, closely matching the true residual patterns. Furthermore, the violin plots in Figure 2 confirm that LSCP maintains superior performance consistency compared to baseline methods.

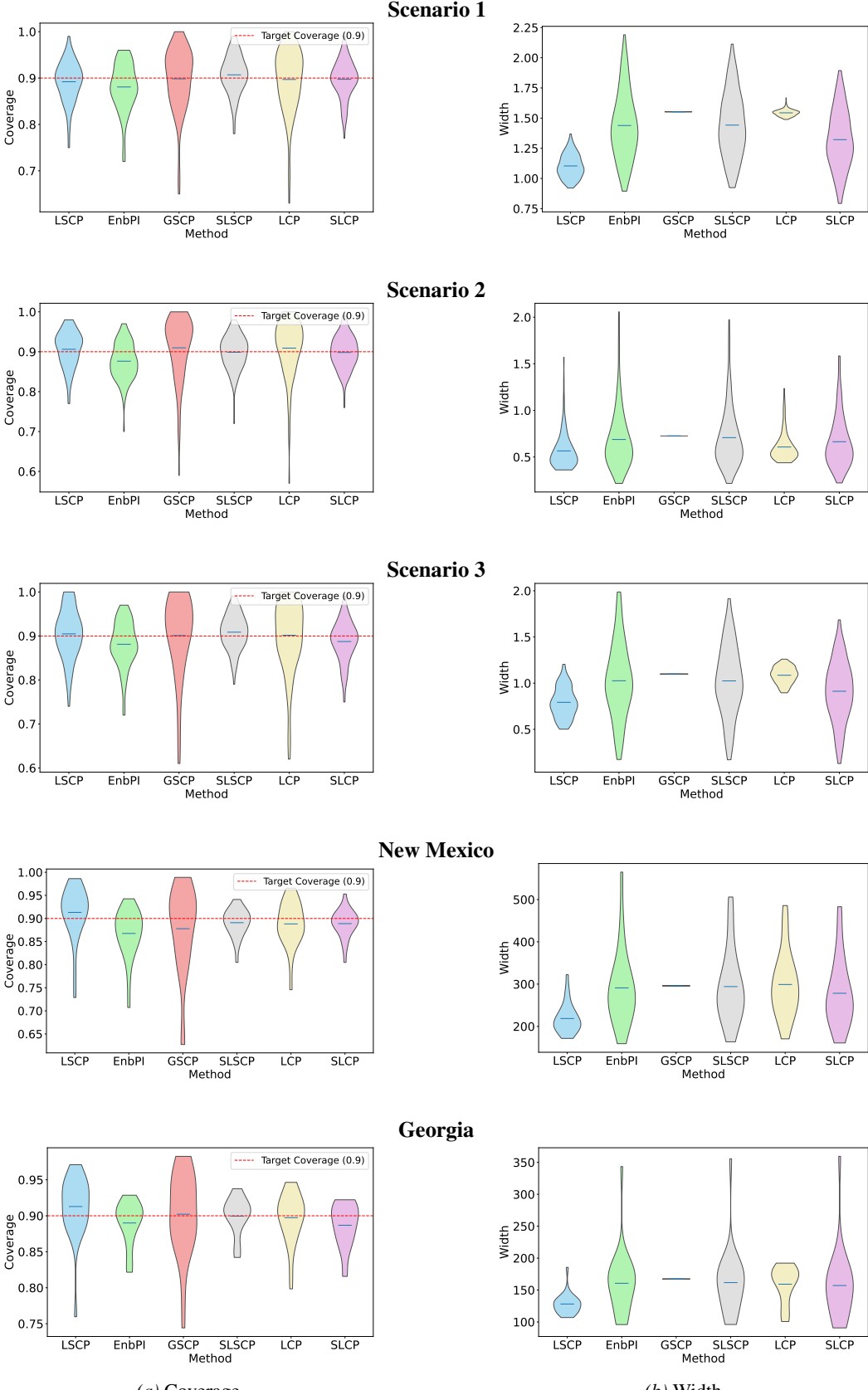

*(a)* Coverage         *(b)* Width

*Figure 2.* The violin plots on the left show the distribution of coverage across different areas, while the plots on the right show the distribution of width. Each row represents a different scenario or location.

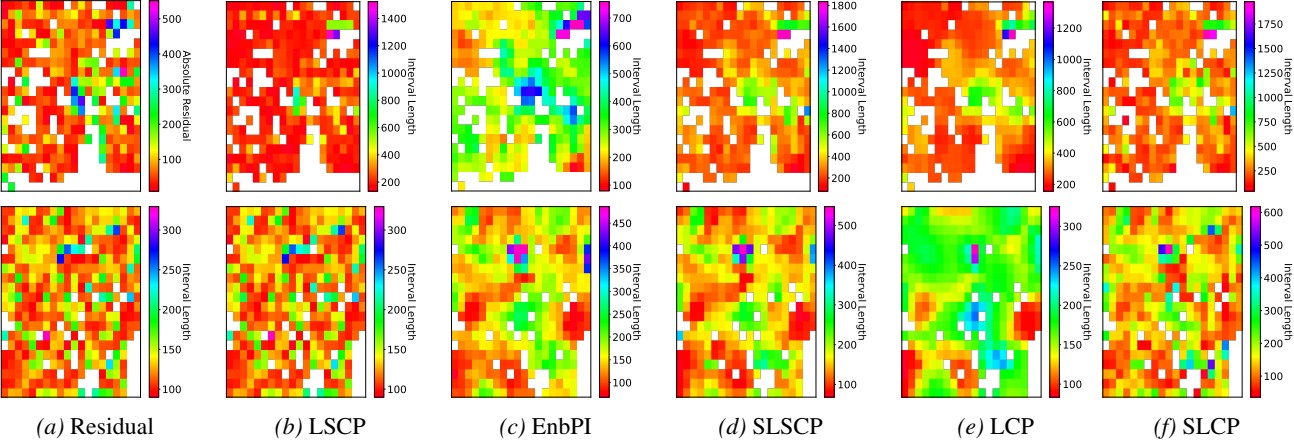

*Figure 3.* The heatmaps illustrate the width of the prediction intervals for each method in the NM (top row) and GA (bottom row) datasets.

