# OpenReview forum: "Spatial Conformal Inference through Localized Quantile Regression"
_ICML.cc/2026/Conference — ICML 2026 regular_

### Official Review · Reviewer_3HoW · 2026-03-09

**Soundness:** 3
**Presentation:** 2
**Significance:** 3
**Originality:** 2
**Overall Recommendation:** 4
**Confidence:** 2

**Summary:**

This paper proposes Localized Spatial Conformal Prediction (LSCP), a method for constructing prediction intervals for spatial data with dependence. Standard conformal prediction relies on exchangeability assumptions, which are violated in spatial settings due to spatial dependence and heterogeneity. The proposed approach addresses this issue by localizing the conformal calibration step. Specifically, the method constructs a feature vector from residuals of spatially neighboring points and applies quantile regression (implemented using Quantile Random Forests) to estimate local residual quantiles, which are then used to construct prediction intervals. The authors provide theoretical analysis showing that under spatial mixing and stationarity assumptions, the proposed method achieves asymptotic conditional coverage. Empirical evaluations on synthetic spatial data and several real-world datasets demonstrate that LSCP achieves target coverage while producing narrower prediction intervals compared to several baseline conformal methods.

**Compliance With Llm Reviewing Policy:**

Affirmed.

**Final Justification:**

Since the authors have shared the results of additional experiments, which have further clarified the effectiveness of the method, I am changing the rating to “Weak Accept 4.”

**Key Questions For Authors:**

See Weaknesses.

**Limitations:**

Yes

**Strengths And Weaknesses:**

Strengths
- The paper addresses uncertainty quantification for spatial data under dependence, which is a practically relevant and non-trivial extension of conformal prediction methods. Handling spatial dependence while maintaining coverage guarantees is an important challenge.
- The proposed method is relatively simple and can be applied as a wrapper around arbitrary prediction models. The use of localized residual information is intuitive and potentially useful in heterogeneous spatial environments.
- The paper provides both theoretical analysis and empirical validation. The theoretical results establish asymptotic coverage guarantees under spatial mixing conditions, while the experiments demonstrate empirical coverage and interval width improvements.
- The authors evaluate the method on both synthetic spatial processes and several real-world datasets, which helps illustrate the behavior of the method in different settings.

Weaknesses
- The core idea—localizing conformal calibration using nearby observations—is closely related to existing localized or weighted conformal prediction approaches. The primary novelty appears to lie in the specific residual-based feature construction and its application to spatial data, which may be viewed as an incremental extension.
- The theoretical results assume a consistent conditional quantile estimator under spatial mixing conditions, while the practical implementation relies on Quantile Random Forests. It is not clear whether the assumptions required for the theoretical guarantees hold for this estimator in practice.
- All experiments use a KNN regressor as the base prediction model. It is unclear whether the reported improvements would persist when using stronger spatial predictors (e.g., neural networks). The improvements may partly depend on the choice of the base model.
- The method relies on k-nearest spatial neighbors to characterize the local residual distribution. This implicitly assumes spatial smoothness of the residual distribution, which may not hold in the presence of spatial discontinuities or sharp boundaries. The sensitivity to the choice of k is not deeply analyzed.

---

> ### Author Rebuttal · Authors · 2026-03-30
>
> We sincerely thank the reviewer for the detailed and thoughtful feedback. Below, we provide our responses to the questions.
>
> **Wk 1: The novelty is an incremental extension**
>
> We agree that LSCP belongs to the broader family of localized and weighted conformal methods, and we will revise the paper to make this positioning clearer. That said, we believe the contribution is more than an incremental spatial extension. Algorithmically, **LSCP is a plug in framework for estimating local residual quantiles**: QRF is only one efficient implementation, other quantile estimators can also be used, and the empirical quantile recovers a localized GSCP special case. Its key difference from SLSCP and LCP is that it replaces fixed user specified kernels with data adaptive weights learned from neighborhood residual features, allowing it to capture richer local dependence than hand designed kernels based only on spatial distance or feature similarity.
>
> The theoretical contribution is also broader than this implementation choice. Instead of assuming exchangeability or i.i.d., we are the **first to analyze weighted empirical quantiles under stationary spatial mixing** and establish a finite sample coverage gap bound together with asymptotic conditional coverage. Appendix A further shows that the proof is formulated at the level of weighted empirical distributions rather than a specific forest architecture, with QRF serving only as one convenient estimator that produces a weighted empirical CDF with learned weights.
>
> **Wk 2: Theory assumes a consistent quantile estimator**
>
> We would like first to clarify that the explicit estimation quality assumption in our theory is **not a consistency requirement on the quantile estimator but on the base predictor**. The quantile stage enters separately through the weighted empirical quantile formulation and the weight decay condition in Assumption 4.1, which the paper notes is automatically satisfied by QRF under a standard minimum leaf size condition in practice. In this sense, the quantile regression step does not introduce a separate misspecification term in our coverage bound. Rather, it mainly improves interval width by its data adaptivity. Besides, many standard quantile estimators are known to be asymptotically consistent under suitable regularity conditions (Meinshausen 2006,Steinwart 2011), which includes quantile random forests.
>
> **Wk 3: Base model is limited to KNN**
>
> The following are additional results on MLP and random forest, which shows that our method is still significantly better even under stronger predictors.
>
> ### MLP
>
> | Method | S1 Width | S1 Cov | S2 Width | S2 Cov | S3 Width | S3 Cov | GA Width | GA Cov | NM Width | NM Cov |
> | :--- | :---: | :---: | :---: | :---: | :---: | :---: | :---: | :---: | :---: | :---: |
> | **LSCP** | 1.18 | 0.91 | 0.51 | 0.92 | 0.83 | 0.92 | 102.30 | 0.90 | 170.48 | 0.92 |
> | **EnbPI** | 1.33 | 0.88 | 0.59 | 0.87 | 0.89 | 0.87 | 114.79 | 0.88 | 182.36 | 0.89 |
> | **GSCP** | 1.45 | 0.90 | 0.66 | 0.91 | 1.03 | 0.90 | 117.54 | 0.90 | 194.29 | 0.90 |
> | **SLSCP** | 1.29 | 0.89 | 0.65 | 0.89 | 0.95 | 0.89 | 116.99 | 0.89 | 185.64 | 0.90 |
> | **LCP** | 1.45 | 0.90 | 0.66 | 0.91 | 1.03 | 0.90 | 117.21 | 0.89 | 182.57 | 0.89 |
>
> ---
>
> ### Random Forest
>
> | Method | S1 Width | S1 Cov | S2 Width | S2 Cov | S3 Width | S3 Cov | GA Width | GA Cov | NM Width | NM Cov |
> | :--- | :---: | :---: | :---: | :---: | :---: | :---: | :---: | :---: | :---: | :---: |
> | **LSCP** | 0.89 | 0.91 | 0.47 | 0.90 | 0.64 | 0.92 | 124.44 | 0.91 | 199.72 | 0.93 |
> | **EnbPI** | 1.26 | 0.90 | 0.56 | 0.87 | 0.89 | 0.90 | 162.64 | 0.89 | 266.92 | 0.89 |
> | **GSCP** | 1.45 | 0.90 | 0.64 | 0.91 | 1.09 | 0.90 | 165.56 | 0.90 | 293.07 | 0.91 |
> | **SLSCP** | 1.32 | 0.92 | 0.61 | 0.89 | 0.94 | 0.91 | 164.83 | 0.89 | 269.31 | 0.90 |
> | **LCP** | 1.46 | 0.90 | 0.64 | 0.91 | 1.09 | 0.90 | 163.85 | 0.90 | 261.34 | 0.90 |
>
> **Wk 4:Rely on spatial smoothness**
>
> We agree that methods based only on distance smoothing can struggle at sharp spatial boundaries. LSCP mitigates this by using geographic neighbors only to construct a local residual representation, then learning data adaptive weights from those residual patterns rather than from distance alone. This lets the method borrow strength from points with similar local uncertainty patterns, even when they are not the closest geographically. Sensitivity to k is already studied in Appendix C.6: coverage is highly stable across different choices of k on both New Mexico (0.923 to 0.926) and Georgia (0.905 to 0.909), with only minor width changes, indicating high robustness. Empirically, LSCP also tracks sharp transitions and heteroskedastic regions better than smoother baselines, and on sparse, imbalanced New Mexico dataset, it remains closest to the target area level coverage while keeping tighter intervals.
>
> 1.Meinshausen, N.(2006)Quantile regression forests.
>
> 2.Steinwart, I.(2011)Estimating conditional quantiles with the help of the pinball loss.

---

> > ### Author Rebuttal · Reviewer_3HoW · 2026-04-02
> >
> > Thank you for your detailed response. Since the authors have shared the results of additional experiments, which have further clarified the effectiveness of the method, I am changing the rating to “Weak Accept 4.”

---

### Official Review · Reviewer_PQtv · 2026-03-12

**Soundness:** 4
**Presentation:** 4
**Significance:** 3
**Originality:** 3
**Overall Recommendation:** 5
**Confidence:** 3

**Summary:**

Localized Spatial Conformal Prediction (LSCP) couples local quantile regression with conformal calibration on spatial neighborhoods to capture local heterogeneity. By applying quantile regression to neighborhood residuals, the method learns data-adaptive localization and successfully adapts to complex spatial dependencies.

LSCP ensures finite-sample marginal coverage under spatial exchangeability and achieves asymptotic conditional coverage under stationarity and spatial mixing. Notably, the theoretical framework establishes finite-sample bounds on the coverage gap without exchangeability, only with the help of stationary and spatial mixing on the noisy terms.

**Compliance With Llm Reviewing Policy:**

Affirmed.

**Final Justification:**

Fully resolved my concerns.

**Key Questions For Authors:**

1. To what extent are the spatial locations of the testing sets contained within the geographic bounds (e.g., the convex hull) of the training and calibration sets? How does the method perform strictly extrapolated test points?
2. Could the authors discuss the potential for employing a data-adaptive neighborhood size selection method (e.g., using a fixed physical radius rather than a fixed neighbor count $k$) to better handle severe density variations?
3. While Appendix C mentions using 50 points for locality-based methods, could you clarify if this neighborhood size ($k$) was universally fixed across all real-world datasets, or if it was explicitly tuned per dataset based on density?

**Limitations:**

yes

**Strengths And Weaknesses:**

Strengths
1. The empirical results clearly demonstrate that LSCP consistently outperforms baseline methods, particularly in Scenarios 2 and 3 where the noise is nonlinear and heteroskedastic. This highlights the method's strong practical performance even when pushed beyond the explicit theoretical assumptions of stationary processes with sufficient spatial mixing (Assumption 4.3).
2. The approach effectively utilizes data-adaptive weights to leverage residual dependencies, yielding prediction intervals that are as narrow as possible while rigorously maintaining coverage.
3. The theoretical proofs are sound, effectively bounding the distance between the estimated conditional CDF of the residuals and the true CDF of the noise by explicitly conditioning on the test feature $X_{n+1}$ and location $s_{n+1}$. This rigorous approach yields a well-defined bounded error for the conditional coverage. Also, the authors successfully substitute the standard exchangeability assumption with stationarity and spatial mixing to establish robust coverage guarantees.

Weakness
1. The reliance on a fixed neighborhood size ($k$) is a significant limitation. While a fixed $k$ performs well in dense urban areas, it may necessitate an excessively large geographic radius in sparse rural settings (such as the New Mexico dataset) to capture a sufficient number of informative neighbors. This structural limitation potentially hinders the method's real-world applicability across heterogeneous geographies.
2. A critical failure mode arises if a test point is located far outside the geographic bounds of the calibration set. In such extrapolation scenarios, the Quantile Random Forest (QRF) may output unpredictable or flat weights, potentially causing the prediction intervals to break down.

---

> ### Author Rebuttal · Authors · 2026-03-30
>
> We sincerely thank the reviewer for the detailed and thoughtful feedback. Below, we provide our responses to the questions.
>
> **Wk1: Rely on fixed neighborhood size**
>
> We agree that a fixed neighborhood size can be challenging in heterogeneous geographies, especially in sparse regions. However, LSCP is less restricted by this than methods based only on distance smoothing. After the neighborhood is formed, LSCP uses it to construct a local residual representation, and the quantile estimator then learns data-adaptive weights from these residual patterns rather than relying only on geographic distance. This allows the method to borrow strength from calibration points with similar local uncertainty structure, even if they are not the closest geographically.
>
> Besides, the sensitivity analysis of neighborhood size k in Appendix C.6 shows that the empirical coverage is very stable on both New Mexico and Georgia with only minor changes in interval width. That means LSCP is very robust to the change of k. At the same time, we do not claim that fixed k fully resolves the challenge of extremely sparse regions. Rather, the empirical evidence shows that even in such settings, LSCP remains closest to the target coverage while yielding tighter intervals than the baselines.
>
> **Wk2: Extrapolation scenarios**
>
> First, we would like to clarify that our coverage guarantee does not depend on the quality of a particular quantile estimator such as QRF. The theory is stated for weighted quantile estimators under the spatial mixing setting, rather than for one specific regression model. Therefore, rare extrapolation cases do not by themselves invalidate the overall coverage guarantee, since coverage is an aggregate quantity. Of course, if the extrapolated test points come from a residual distribution that is substantially different from that of the calibration data, then the assumptions underlying conformal prediction are no longer satisfied. In that case, coverage may deteriorate for any conformal method, and this issue is not specific to QRF.
>
> Second, we believe LSCP is less restricted by geography than existing localized approaches. Rather than relying only on spatial proximity, LSCP learns from local residual patterns, which define a richer representation than geographic distance alone. This allows the quantile estimator to borrow strength from locations that may be far apart geographically but share similar local error structure, leading to more robust and adaptive prediction intervals.
>
> **Q1: Performance on extrapolated points**
>
> In real data experiments, almost all test locations lie within the convex hull of the training and calibration locations. As discussed in our response to Weakness 2, QRF itself does not introduce an additional misspecification term into the coverage gap. Coverage deteriorates only when the underlying data assumption is violated, namely when the residual distribution at extrapolated test points differs substantially from that of the calibration data. In that case, the issue reflects a breakdown of the assumptions required by conformal prediction more broadly. The following results demonstrate a boundary-case analysis on the New Mexico dataset. By restricting the test set to boundary data points, LSCP demonstrates reliable coverage and efficiency.
> | Method | Wid |  Cov |
> | :--- | :---: | :---: |
> | **LSCP** | 121.21 | 0.90 |
> | **EnbPI** | 151.40 | 0.89 |
> | **GSCP** | 155.26 | 0.89 |
> | **SLSCP** | 152.58 | 0.90 |
> | **LCP** | 150.35 | 0.89 |
>
> **Q2: Neighborhood selection**
>
> We thank the reviewer for this suggestion. We agree that there are multiple possible ways to define neighborhoods, and that data adaptive choices may be especially useful under strong density variation. In the current paper, we use a fixed neighbor count mainly because the neighborhood residuals are used to construct the residual feature, which requires a fixed input dimension across locations. This makes the quantile estimation step simple and stable in practice. That said, the neighborhood need not be defined only through geographic distance. One could also form neighborhoods using nearest neighbors in a more informative feature space, which may better handle severe density variation or heterogeneous spatial structure. We chose geographic distance in the current work because it is simple, intuitive, and easy to interpret, but we agree that more adaptive neighborhood constructions could further improve performance depending on the dataset.
>
> **Q3: Neighborhood size**
>
> In the current experiments, the neighborhood size was fixed rather than tuned per dataset density. We used 50 neighbors for all real world datasets, except for the Farming dataset, where we used 5 because the dataset contains only 500 total samples. We will clarify this explicitly in the appendix. We agree that density adaptive neighborhood selection is a promising direction, although Appendix C.6 suggests that LSCP is fairly robust to the choice of neighborhood size.

---

> > ### Author Rebuttal · Reviewer_PQtv · 2026-04-03
> >
> > All concerns are resolved.

---

### Official Review · Reviewer_qA9a · 2026-03-12

**Soundness:** 3
**Presentation:** 3
**Significance:** 2
**Originality:** 2
**Overall Recommendation:** 4
**Confidence:** 3

**Summary:**

This paper proposes a conformal prediction method called Localized Spatial Conformal Prediction (LSCP), which produces confidence intervals for new spatial locations. It is challenging to quantify uncertainty via conformal prediction in the spatial settiing due to the lack of exchangeability and natural indexing. The main theoretical result establishes finite-sample conditional coverage under regularity conditions, with asymptotic and marginal coverage guarantees derived as corollaries. Empirical evaluations on both simulated and real datasets show that LSCP produces significantly narrower prediction intervals while maintaining the desired coverage rate.

**Compliance With Llm Reviewing Policy:**

Affirmed.

**Key Questions For Authors:**

- Q1. Experimental settings in quantile random forest

This article employs QRF for localized quantile regression. However, the discrepancy of numbers of neighbors between LSCP and other baseline models is not fully discussed. Providing these details would strengthen the result. In addition, the theoretical weight decay condition imposes constraints on the minimum number of samples per leaf, and it would be useful to clarify how this aligns with the actual experimental settings or whether adjustments were made to satisfy the theoretical assumptions.

- Q2. Assumptions on estimation quality

The article notes that the upper bound of the average prediction error, $\delta$, does not need to converge to zero. However, convergence of $\delta$ is required to establish the asymptotic coverage rate. It would be helpful for the authors to clarify this apparent discrepancy and explain more clearly the significance of this remark.

- Q3. Discussion of interval width

In the experiments, the primary advantage of the proposed method appears to be the smaller interval widths. However, the underlying reason for this phenomenon is not fully discussed. It would be useful for the authors to clarify whether this improvement arises from the QRF, the data-adaptive weighting, or simply from the adjustable quantile level used to explicitly minimize interval widths. Such clarification would provide a better understanding of effective strategies for conformal prediction on spatial data.

- Q4. Figures in real-data experiments

There appears to be a discrepancy in the references to plots in the real-data experiments. The captions for Figures 1 and 4 do not seem to match the content of the figures and may instead be referring to plots in the appendix. Clarifying these references would improve the clarity and presentation of the results.

**Limitations:**

No explicit limitations are discussed; the manuscript would benefit from including them.

**Strengths And Weaknesses:**

Soundness: This article is technically sound overall. The theoretical analysis is carefully developed, and the assumptions used to characterize stationarity and spatial mixing are standard. The proof techniques for establishing conditional coverage are also consistent with existing analyses in conformal prediction. The empirical performance of LSCP is compared with several relevant baseline methods in terms of coverage rate and interval width, which are standard evaluation metrics for conformal prediction methods. That said, the asymptotic convergence results may rely on relatively strong assumptions regarding the estimation quality, which could benefit from further justification.

Presentation: The paper is generally well organized and follows a standard structure for presenting a new algorithm. The authors clearly compare the proposed method with related approaches from multiple perspectives, which helps position the contribution within the existing literature. However, some implementation details could be described more clearly. In particular, the configuration of the quantile random forest (QRF) method is not fully specified, especially in relation to the assumptions used in the theoretical analysis. In addition, the optimization procedure used to determine the quantile level that minimizes the interval width is not clearly described, which may make it challenging to fully reproduce the algorithm.

Significance: This article studies uncertainty quantification for spatial data, where the lack of exchangeability makes reliable uncertainty estimation challenging. This is an important problem in spatial prediction. However, the methodological development mainly builds on established ideas, and the potential to inspire substantially new directions may be limited. Nevertheless, the proposed method provides a practical approach for constructing prediction intervals in spatial settings, which may be useful for certain spatial prediction applications.

Originality: The proposed method builds on existing approaches in local conformal prediction, including the use of k-nearest neighbors and quantile estimation. The adaptive weighting scheme is implemented through quantile random forests (QRF), and the adaptivity of wrights is therefore largely inherited from the QRF model itself. As a result, the methodological novelty beyond existing techniques appears relatively limited.

---

> ### Author Rebuttal · Authors · 2026-03-30
>
> We sincerely thank the reviewer for the detailed and thoughtful feedback. Below, we provide our responses to the questions.
>
> **Q1: Experimental settings in quantile random forest**
>
> We thank the reviewer for pointing this out. We will clarify the experimental settings more explicitly in the revision. As stated in Appendix C.4, the base predictor is the same across methods, namely a KNN regressor with k=5. For the locality based methods LSCP, EnbPI, and SLSCP, we use the same neighborhood size of 50, so LSCP does not benefit from a larger local sample than these baselines. Besides, the sensitivity analysis in Appendix C.6 also shows that LSCP is not sensitive to the size of neighborhood and consistently outperforms baseline methods. So the neighborhood size is not a specific choice that favors our method.
>
> Regarding the theoretical weight decay condition, Assumption 4.1 provides a sufficient condition and explicitly notes that it automatically holds for QRF when we set the minimum samples per leaf in a certain way. However, in our implementation, we use minimum samples per leaf to be 1 rather than explicitly enforcing this sufficient condition. We will clarify this more explicitly in the revision. Therefore, the experiments should be interpreted as evaluating LSCP in a general setting rather than the more restrictive one used in the proof, and the favorable empirical results suggest that our method remains robust even when that sufficient condition is not imposed exactly.
>
> **Q2: Assumptions on estimation quality**
>
> We thank the reviewer for highlighting this important point. To begin, we clarify the necessity of this condition. While standard conformal prediction relies on the restrictive assumption of exchangeability, our framework relaxes this requirement, necessitating alternative theoretical assumptions. Furthermore, we aim to prove a much stronger result of conditional coverage, which is known to be impossible to achieve without additional requirements (Barber, 2019). Fundamentally, one cannot expect valid inference for arbitrary algorithms on challenging data, so we require that the base algorithm be sufficiently expressive to capture underlying patterns, such as trends and non-homogeneity. Assumption 4.2 is the specific technical condition required for proving local asymptotic conditional validity, guaranteeing that our intervals are optimally tight in every specific sub-region of the space. Crucially, however, LSCP remains empirically robust even when this assumption does not strictly hold.
>
> From a practical standpoint, Assumption 4.2 is satisfied in standard machine learning scenarios where the base predictor is capable of approximating the underlying signal as the sample size grows. This includes high-dimensional regression using regularized linear models on sparse data (Bickel, 2009) as well as non-parametric approaches like random forests or neural networks applied to sufficiently smooth processes (Chen, 1999).
>
> **Q3: Discussion of interval width**
>
> We thank the reviewer for raising this question. We agree that the improvement in interval width deserves clearer discussion. The main driver is the data adaptive weighting, rather than QRF itself or the optimization over the quantile level. Conceptually, LSCP differs from SLSCP and LCP because it learns the weights used in the local residual quantile from neighborhood residual features, instead of relying on a fixed user-specified kernel. QRF is the computationally efficient estimator used in our implementation, but it is not the core idea of the method. Our paper explicitly notes that other quantile regression estimators could also be used, and Appendix A shows that the QRF output is itself a weighted empirical quantile over calibration residuals. Empirically, this learned weighting appears to be what allows LSCP to track sharp transitions and heteroskedastic regions more closely than the smoother local averaging baselines, which is reflected in the heatmaps, the violin plots, and the consistently smaller widths in both the synthetic and real data experiments while maintaining valid coverage. On the other hand, the optimization over $\beta$ only selects the tightest interval induced by a given estimated quantile function while not creating a better estimate by itself.
>
> **Q4: Figures in real-data experiments**
>
> We thank the reviewer for their careful reading and helpful feedback. We apologize for the confusion regarding the citations. While the captions for Figure 1 and Figure 4 in the main text are correct, we acknowledge that there is in-text reference to 'Figure 1' which intended to point to Figure 1 in the Appendix. We will explicitly clarify these distinctions in the revised manuscript to ensure better navigation between the main body and the supplemental material.
>
> 1. Bickel, P. J. (2009) Simultaneous analysis of Lasso and Dantzig selector
>
> 2. Chen, X. (1999) Improved rates and asymptotic normality for nonparametric neural network estimators

---

> > ### Author Rebuttal · Reviewer_qA9a · 2026-04-01
> >
> > Thanks for the responses. My comments are well addressed and I have updated the score as "4: Weak accept".

---

### Official Review · Reviewer_sd2S · 2026-03-12

**Soundness:** 3
**Presentation:** 3
**Significance:** 3
**Originality:** 3
**Overall Recommendation:** 4
**Confidence:** 4

**Summary:**

The paper studies prediction intervals for spatial data, where the exchangability assumptions underlying standard conformal prediction generally fail. The proposed method first fits a prediction model, then uses nearby calibration residuals to estimate location-specific residual quantiles, which determine the interval width at each test location. The intervals are adaptive because a single quantile regression model uses neighboring residual features to estimate location-specific residual quantiles.The main contribution is theory: rather than exact finite-sample conformal coverage, the paper establishes finite-sample bounds on the coverage gap and asymptotic validity under spatial stationarity and mixing assumptions.

**Compliance With Llm Reviewing Policy:**

Affirmed.

**Final Justification:**

The authors addressed my comment. In particular, I agree that preserving finite-sample guarantees for conformal prediction under spatial correlation is challenging. I have increased my score and now lean toward acceptance.

The method still appears somewhat ad hoc, in that it relies on a specific set of modeling choices and estimation procedures. This limits the paper’s broader theoretical contribution. However, the proposed method is interesting, supported by theoretical results under appropriate conditions, and appears to perform well in practice.

**Key Questions For Authors:**

1. I have some questions about to what degree the procedure is ``local" in space:

-  Is there any reason to expect a single global quantile model to behave locally simply because it uses local features as inputs?

- While regression trees and random forests can localize in feature space, the quantile model here does not appear to use spatial location itself as a feature. Instead, it uses residuals from neighboring points as inputs. If so, does localization in this feature space necessarily translate into localization in geographic space?

- More generally, what ensures that the fitted global quantile model preserves local spatial structure, rather than smoothing across regions with different uncertainty/residual patterns?

2. Fitting a global quantile regression model on local residual features makes the method more sensitive to modeling error. This is also reflected in the theory, which requires consistency of the quantile estimator. A key advantage of classical conformal prediction is that coverage does not depend on the quality of the underlying predictive model. Here, by contrast, if the quantile regression model is misspecified or poorly calibrated, interval coverage may deteriorate. This raises the question of whether the estimated local quantiles should themselves be calibrated, so that they better agree with the empirical local quantiles.

3. Relatedly, would existing localized conformal methods based on local empirical quantiles be more robust in practice? Intuitively, such methods rely more directly on local calibration rather than on fitting an additional global model. More broadly, the method does not seem to perform a post hoc conformal correction of the estimated quantiles in the usual sense. Instead, it uses conformity scores as inputs to learn quantiles from which prediction intervals are constructed. As a result, beyond its use of conformity scores, the connection to conformal prediction seems weaker than in standard conformal methods.

**Limitations:**

Yes

**Strengths And Weaknesses:**

# Strengths

1. The authors tackle the challenging problem of constructing prediction intervals for spatially correlated data. Addressing this setting requires accounting for spatial dependence, stationarity, and mixing, which makes the problem substantially more difficult than the standard exchangeable setting.

2. The proposed method is conceptually simple and modular: it combines conformal prediction with localized quantile estimation, making it compatible with a wide range of underlying predictive models.

3. The paper provides a theoretical analysis for the spatially dependent setting, establishing bounds on conditional coverage error and asymptotic validity under spatial mixing assumptions.



# Weaknesses

1. A limitation is that the localized method does not retain the exact finite-sample coverage guarantee that makes classical conformal prediction especially attractive. Instead, the paper provides a finite-sample bound on the conditional coverage gap and relies on asymptotic arguments for validity under assumptions such as spatial mixing and estimation consistency.

2. Methodologically, as summarized in Table 1, the proposed method can be viewed as a spatial adaptation of existing localized conformal ideas rather than a fundamentally new conformal construction. Its main methodological difference is that it replaces empirical quantiles of nearby residuals with location-specific quantiles predicted by a single global quantile random forest fit on neighboring residual features, thereby inducing data-adaptive weights. This additional modeling step may require a sufficiently large calibration set. The main novelty therefore appears to lie in extending the coverage analysis to spatially dependent data. The paper’s literature review also identifies Guan (2023) and kernel-weighted spatial conformal methods as closely related.


Guan, Leying. "Localized conformal prediction: A generalized inference framework for conformal prediction." Biometrika 110.1 (2023): 33-50.

---

> ### Author Rebuttal · Authors · 2026-03-30
>
> We sincerely thank the reviewer for the detailed and thoughtful feedback. Below, we provide our point-by-point responses to the raised concerns and questions.
>
> **Weakness 1: LSCP does not retain exact finite-sample coverage.**
>
> We agree that the **learned-weight version of LSCP** does not claim the same exact finite-sample marginal guarantee as classical split conformal under exchangeability. This is the intended scope: LSCP provides **finite-sample bounds on the coverage gap** and **asymptotic conditional coverage** under stationarity and spatial mixing. Exact marginal coverage is recovered in the exchangeable special case when the method reduces to GSCP. In spatially localized settings, the exchangeability-based argument is generally **unavailable** once one localizes quantile estimation. As shown in **Appendix B**, localizing improves adaptivity but breaks the standard exchangeability argument. Our contribution provides **meaningful coverage control in non-exchangeable spatial regimes** with explicit coverage-gap bounds.
>
> **Weakness 2: Novelty and calibration set size.**
>
> LSCP is part of the **weighted conformal family**, but replaces fixed kernels with **data-adaptive weights** learned from neighborhood residual features. It is a **plug-in framework**. While we use QRF, other estimators apply. **Appendix A** shows the QRF estimator is a **weighted empirical quantile**, recovering localized GSCP as a special case. Regarding data requirements, LSCP is effective in small-sample regimes: on the **Farming dataset ($N=500$, $n_{cal}=200$)**, LSCP achieves nominal 90% coverage with the **narrowest intervals** among all baselines.
>
> **Q1: Definition of "Local" and Spatial Structure.**
>
> * **Global model vs. Local behavior:** In our algorithm, each calibration location is represented by a local training pair consisting of its neighborhood residual vector and its corresponding residual, and the quantile model is fit globally. Since the predicted quantile can be interpreted as a weighted empirical quantile, the model effectively assigns greater weight to calibration points whose local residual patterns are most similar to those of the target location. In this sense, LSCP does not impose a single global notion of uncertainty across space, but instead borrows information from locations with similar local structure.
>
> * **Geographic vs. Feature localization:** Inputs are **geographically grounded** (residuals ordered by spatial distance). Nearby locations induce similar representations, but the model can borrow strength from distant locations with comparable uncertainty patterns. This captures richer dependence than fixed-distance kernels.
>
> * **Oversmoothing:** Smoothing is mitigated by training on **distance-ordered neighborhood vectors**. Using a weighted empirical CDF (Appendix A), the model targets similar local residual structures rather than averaging uniformly across regions.
>
> **Q2: Sensitivity to Modeling Error.**
>
> Our theory does not require a standalone consistency property for the quantile regressor. **Assumption 4.2** is on the base predictor. We use this to bound the discrepancy between the weighted empirical distribution and the true noise. The quantile stage enters via the **weight decay condition (Assumption 4.1)**, which QRF automatically satisfies under a standard minimum leaf size condition. In this sense, the quantile regression step does not introduce a separate misspecification term in our coverage bound. Rather, it mainly improves interval width by its data adaptivity. Moreover, many standard quantile estimators are known to be asymptotically consistent under suitable regularity conditions (Meinshausen 2006, Takeuchi 2006, Steinwart 2011), and this includes quantile random forests.
>
> **Q3: Robustness and Connection to Conformal Prediction.**
>
> Empirically, methods based on fixed empirical quantiles are not more robust when there is heterogeneity in data. In tables and violin plots, LSCP maintains nominal coverage with **lower or comparable variance**, which suggests a consistent superior performance. This is because LSCP learns local residual patterns globally.
>
> On the other hand, the connection to conformal prediction remains direct. LSCP starts from calibrating nonconformity scores, defines the prediction interval through a quantile of those scores, and reduces to GSCP when the empirical quantile is used. Appendix A shows that the QRF is itself a weighted empirical quantile over calibration residuals, replacing hand-specified kernel weights with learned ones. In this sense, LSCP is closely tied to the weighted and localized conformal framework while extending it from fixed kernel weighting to learned data adaptive weighting.
>
> 1. Meinshausen, N. (2006). Quantile regression forests.
> 2. Takeuchi, I. et al. (2006). Nonparametric quantile estimation.
> 3. Steinwart, I. & Christmann, A. (2011). Estimating conditional quantiles with the help of the pinball loss.

---

> > ### Author Rebuttal · Reviewer_sd2S · 2026-04-03
> >
> > The authors addressed my comment. In particular, I agree that preserving finite-sample guarantees for conformal prediction under spatial correlation is challenging. I have increased my score and now lean toward acceptance.
> >
> > My main reservation is that method still appears somewhat ad hoc, in that it relies on a specific set of modeling choices and estimation procedures. This limits the paper’s broader theoretical contribution. However, the proposed method is interesting, supported by theoretical results under appropriate conditions, and appears to perform well in practice.

---

### Decision · Program_Chairs · 2026-04-30

**Decision:**

Accept (regular)

**Comment:**

This submission addresses an important and technically challenging problem: constructing reliable prediction intervals for spatially dependent data. Reviewers agreed that LSCP is a simple, modular method backed by meaningful theory and strong empirical results showing near-nominal coverage with consistently tighter intervals than relevant baselines. The main weaknesses raised across reviews were that the method appears somewhat incremental relative to prior localized or weighted conformal approaches, that some guarantees rely on nontrivial assumptions and a particular implementation path, and that the initial submission left open questions about locality, neighborhood selection, extrapolation behavior, and several implementation details. In the discussion, reviewers focused on whether the global quantile model truly preserves local spatial structure, how much the theoretical guarantees depend on QRF and estimation assumptions, and whether the observed gains in interval width come primarily from adaptive weighting rather than tuning choices or the specific base predictor. The authors’ rebuttal directly addressed these concerns by clarifying the weighted-empirical-quantile interpretation of LSCP, explaining that the key estimation requirement is on the base predictor rather than a separate QRF consistency assumption, and providing additional evidence on neighborhood sensitivity, boundary cases, and stronger base models. Overall, the paper’s novelty is quite limited, so for me this is exactly on the borderline.